# Systematic integration of biomedical knowledge prioritizes drugs for repurposing

**Daniel Scott Himmelstein[1,2], Antoine Lizee[3,4], Christine Hessler[3], Leo Brueggeman[3,5], Sabrina L Chen[3,6], Dexter Hadley[7,8], Ari Green[3], Pouya Khankhanian[3,9], Sergio E Baranzini[1,3]***

[1]Biological and Medical Informatics Program, University of California, San Francisco, San Francisco, United States; [2]Department of Systems Pharmacology and Translational Therapeutics, University of Pennsylvania, Philadelphia, United States; [3]Department of Neurology, University of California, San Francisco, San Francisco, United States; [4]ITUN-CRTI-UMR 1064 Inserm, University of Nantes, Nantes, France; [5]University of Iowa, Iowa City, United States; [6]Johns Hopkins University, Baltimore, United States; [7]Department of Pediatrics, University of California, San Fransisco , San Fransisco, United States; [8]Institute for Computational Health Sciences, University of California, San Francisco, San Francisco, United States; [9]Center for Neuroengineering and Therapeutics, University of Pennsylvania, Philadelphia, United States

**Abstract** The ability to computationally predict whether a compound treats a disease would improve the economy and success rate of drug approval. This study describes Project Rephetio to systematically model drug efficacy based on 755 existing treatments. First, we constructed Hetionet (neo4j.het.io), an integrative network encoding knowledge from millions of biomedical studies. Hetionet v1.0 consists of 47,031 nodes of 11 types and 2,250,197 relationships of 24 types. Data were integrated from 29 public resources to connect compounds, diseases, genes, anatomies, pathways, biological processes, molecular functions, cellular components, pharmacologic classes, side effects, and symptoms. Next, we identified network patterns that distinguish treatments from non-treatments. Then, we predicted the probability of treatment for 209,168 compound–disease pairs (het.io/repurpose). Our predictions validated on two external sets of treatment and provided pharmacological insights on epilepsy, suggesting they will help prioritize drug repurposing candidates. This study was entirely open and received realtime feedback from 40 community members.
DOI: https://doi.org/10.7554/eLife.26726.001

*For correspondence:
sergio.baranzini@ucsf.edu

**Competing interests:** The authors declare that no competing interests exist.

## Introduction

The cost of developing a new therapeutic drug has been estimated at 1.4 billion dollars (*DiMasi et al., 2016*), the process typically takes 15 years from lead compound to market (*Reichert, 2003*), and the likelihood of success is stunningly low (*Hay et al., 2014*). Strikingly, the costs have been doubling every 9 years since 1970, a sort of inverse Moore's law, which is far from an optimal strategy from both a business and public health perspective (*Scannell et al., 2012*). Drug repurposing — identifying novel uses for existing therapeutics — can drastically reduce the duration, failure rates, and costs of approval (*Ashburn and Thor, 2004*). These benefits stem from the rich

**eLife digest** Of all the data in the world today, 90% was created in the last two years. However, taking advantage of this data in order to advance our knowledge is restricted by how quickly we can access it and analyze it in a proper context.

In biomedical research, data is largely fragmented and stored in databases that typically do not "talk" to each other, thus hampering progress. One particular problem in medicine today is that the process of making a new therapeutic drug from scratch is incredibly expensive and inefficient, making it a risky business. Given the low success rate in drug discovery, there is an economic incentive in trying to repurpose an existing drug that has already been shown to be safe and effective towards a new disease or condition.

Himmelstein et al. used a computational approach to analyze 50,000 data points – including drugs, diseases, genes and symptoms – from 19 different public databases. This approach made it possible to create more than two million relationships among the data points, which could be used to develop models that predict which drugs currently in use by doctors might be best suited to treat any of 136 common diseases. For example, Himmelstein et al. identified specific drugs currently used to treat depression and alcoholism that could be repurposed to treat smoking addition and epilepsy.

These findings provide a new and powerful way to study drug repurposing. While this work was exclusively performed with public data, an expanded and potentially stronger set of predictions could be obtained if data owned by pharmaceutical companies were incorporated. Additional studies will be needed to test the predictions made by the models.

DOI: https://doi.org/10.7554/eLife.26726.002

preexisting information on approved drugs, including extensive toxicology profiling performed during development, preclinical models, clinical trials, and postmarketing surveillance.

Drug repurposing is poised to become more efficient as mining of electronic health records (EHRs) to retrospectively assess the effect of drugs gains feasibility (*Wang et al., 2015*; *Xu et al., 2015*; *Brilliant et al., 2016*; *Tatonetti et al., 2012*). However, systematic approaches to repurpose drugs based on mining EHRs alone will likely lack power due to multiple testing. Similar to the approach followed to increase the power of genome-wide association studies (GWAS) (*Stephens and Balding, 2009*; *Sawcer, 2008*), integration of biological knowledge to prioritize drug repurposing will help overcome limited EHR sample size and data quality.

In addition to repurposing, several other paradigm shifts in drug development have been proposed to improve efficiency. Since small molecules tend to bind to many targets, polypharmacology aims to find synergy in the multiple effects of a drug (*Roth et al., 2004*). Network pharmacology assumes diseases consist of a multitude of molecular alterations resulting in a robust disease state. Network pharmacology seeks to uncover multiple points of intervention into a specific pathophysiological state that together rehabilitate an otherwise resilient disease process (*Hopkins, 2008*; *Hopkins, 2007*). Although target-centric drug discovery has dominated the field for decades, phenotypic screens have more recently resulted in a comparatively higher number of first-in-class small molecules (*Swinney and Anthony, 2011*). Recent technological advances have enabled a new paradigm in which mid- to high-throughput assessment of intermediate phenotypes, such as the molecular response to drugs, is replacing the classic target discovery approach (*Iskar et al., 2012*; *Lamb, 2007*; *Qu and Rajpal, 2012*). Furthermore, integration of multiple channels of evidence, particularly diverse types of data, can overcome the limitations and weak performance inherent to data of a single domain (*Hodos et al., 2016*). Modern computational approaches offer a convenient platform to tie these developments together as the reduced cost and increased velocity of in silico experimentation massively lowers the barriers to entry and price of failure (*Hurle et al., 2013*; *Liu et al., 2013*).

Hetnets (short for heterogeneous networks) are networks with multiple types of nodes and relationships. They offer an intuitive, versatile, and powerful structure for data integration by aggregating graphs for each relationship type onto common nodes. In this study, we developed a hetnet (Hetionet v1.0) by integrating knowledge and experimental findings from decades of biomedical research spanning millions of publications. We adapted an algorithm originally developed for social

network analysis and applied it to Hetionet v1.0 to identify patterns of efficacy and predict new uses for drugs. The algorithm performs edge prediction through a machine learning framework that accommodates the breadth and depth of information contained in Hetionet v1.0 (*Himmelstein and Baranzini, 2015a*; *Sun et al., 2011*). Our approach represents an in silico implementation of network pharmacology that natively incorporates polypharmacology and high-throughput phenotypic screening.

One fundamental characteristic of our method is that it learns and evaluates itself on existing medical indications (i.e. a 'gold standard'). Next, we introduce previous approaches that also performed comprehensive evaluation on existing treatments. A 2011 study, named PREDICT, compiled 1933 treatments between 593 drugs and 313 diseases (*Gottlieb et al., 2011*). Starting from the premise that similar drugs treat similar diseases, PREDICT trained a classifier that incorporates five types of drug-drug and two types of disease-disease similarity. A 2014 study compiled 890 treatments between 152 drugs and 145 diseases with transcriptional signatures (*Cheng et al., 2014*). The authors found that compounds triggering an opposing transcriptional response to the disease were more likely to be treatments, although this effect was weak and limited to cancers. A 2016 study compiled 402 treatments between 238 drugs and 78 diseases and used a single proximity score — the average shortest path distance between a drug's targets and disease's associated proteins on the interactome — as a classifier (*Guney et al., 2016*).

We build on these successes by creating a framework for incorporating the effects of any biological relationship into the prediction of whether a drug treats a disease. By doing this, we were able to capture a multitude of effects that have been suggested as influential for drug repurposing including drug-drug similarity (*Gottlieb et al., 2011*; *Li and Lu, 2012*), disease-disease similarity (*Gottlieb et al., 2011*; *Chiang and Butte, 2009*), transcriptional signatures (*Lamb, 2007*; *Qu and Rajpal, 2012*; *Cheng et al., 2014*; *Lamb et al., 2006*; *Iorio et al., 2013*), protein interactions (*Guney et al., 2016*), genetic association (*Nelson et al., 2015*; *Sanseau et al., 2012*), drug side effects (*Campillos et al., 2008*; *Nugent et al., 2016*), disease symptoms (*Zhou et al., 2014*), and molecular pathways (*Pratanwanich and Lió, 2014*). Our ability to create such an integrative model of drug efficacy relies on the hetnet data structure to unite diverse information. On Hetionet v1.0, our algorithm learns which types of compound–disease paths discriminate treatments from non-treatments in order to predict the probability that a compound treats a disease.

We refer to this study as Project Rephetio (pronounced as rep-*het-ee*-oh). Both Rephetio and Hetionet are portmanteaus combining the words repurpose, heterogeneous, and network with the URL het.**io**.

## Results

### Hetionet v1.0

We obtained and integrated data from 29 publicly available resources to create Hetionet v1.0 (*Figure 1*). The hetnet contains 47,031 nodes of 11 types (*Table 1*) and 2,250,197 relationships of 24 types (*Table 2*). The nodes consist of 1552 small molecule compounds and 137 complex diseases, as well as genes, anatomies, pathways, biological processes, molecular functions, cellular components, perturbations, pharmacologic classes, drug side effects, and disease symptoms. The edges represent relationships between these nodes and encompass the collective knowledge produced by millions of studies over the last half century.

For example, *Compound–binds–Gene* edges represent when a compound binds to a protein encoded by a gene. This information has been extracted from the literature by human curators and compiled into databases such as DrugBank, ChEMBL, DrugCentral, and BindingDB. We combined these databases to create 11,571 binding edges between 1389 compounds and 1689 genes. These edges were compiled from 10,646 distinct publications, which Hetionet binding edges reference as an attribute. Binding edges represent a comprehensive catalog constructed from low-throughput experimentation. However, we also integrated findings from high-throughput technologies — many of which have only recently become available. For example, we generated consensus transcriptional signatures for compounds in LINCS L1000 and diseases in STARGEO.

While Hetionet v1.0 is ideally suited for drug repurposing, the network has broader biological applicability. For example, we have prototyped queries for (a) identifying drugs that target a specific

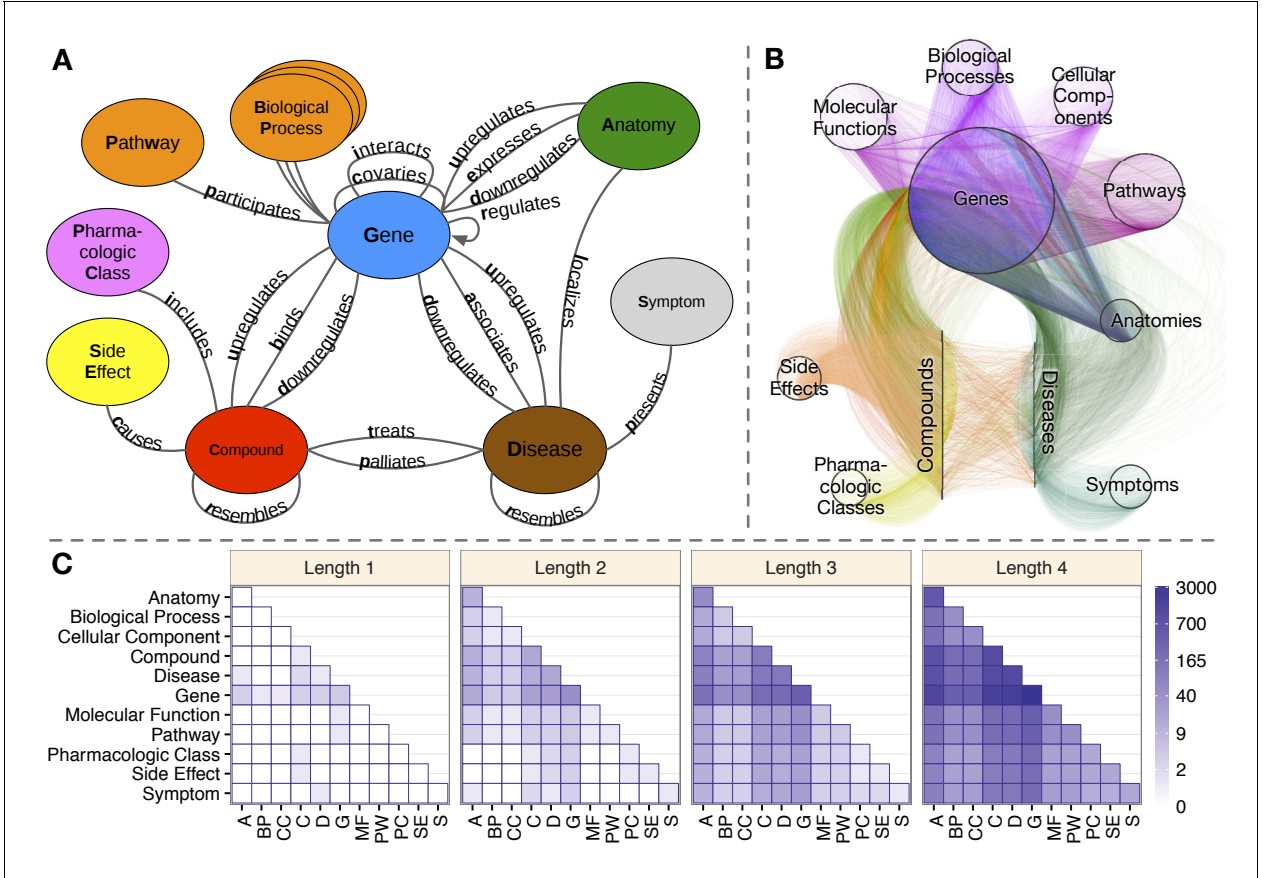

**Figure 1.** Hetionet v1.0. (**A**) The metagraph, a schema of the network types. (**B**) The hetnet visualized. Nodes are drawn as dots and laid out orbitally, thus forming circles. Edges are colored by type. (**C**) Metapath counts by path length. The number of different types of paths of a given length that connect two node types is shown. For example, the top-left tile in the Length 1 panel denotes that Anatomy nodes are not connected to themselves (i.e. no edges connect nodes of this type between themselves). However, the bottom-left tile of the Length 4 panel denotes that 88 types of length-four paths connect Symptom to Anatomy nodes.

DOI: https://doi.org/10.7554/eLife.26726.003

pathway, (b) identifying biological processes involved in a specific disease, (c) identifying the drug targets responsible for causing a specific side effect, and (d) identifying anatomies with transcriptional relevance for a specific disease (**Himmelstein, 2016j**). Each of these queries was simple to write and took less than a second to run on our publicly available Hetionet Browser. Although it is possible that existing services provide much of the aforementioned functionality, they offer less versatility. Hetionet differentiates itself in its ability to flexibly query across multiple domains of information. As a proof of concept, we enhanced the biological process query (b), which identified processes that were enriched for disease-associated genes, using multiple sclerosis (MS) as an example disease. The verbose Cypher code for this query is shown below:

```
MATCH path =
  //Specify the type of path to match
  (n0:Disease)-[e1:ASSOCIATES_DaG]-(n1:Gene)-[:INTERACTS_GiG]-
  (n2:Gene)-[:PARTICIPATES_GpBP]-(n3:BiologicalProcess)
WHERE
  //Specify the source and target nodes
  n0.name = 'multiple sclerosis' AND
  n3.name = 'retina layer formation'
  //Require GWAS support for the Disease-associates-Gene relationship
```

```
AND 'GWAS Catalog' in e1.sources
//Require the interacting gene to be upregulated in a relevant tissue
AND exists((n0)-[:LOCALIZES_DlA]-(:Anatomy)-[:UPREGULATES_AuG]-(n2))
RETURN path
```

The query above identifies genes that interact with MS GWAS-genes. However, interacting genes are discarded unless they are upregulated in an MS-related anatomy (i.e. anatomical structure, e.g. organ or tissue). Then relevant biological processes are identified. Thus, this single query spans four node and five relationship types.

The integrative potential of Hetionet v1.0 is reflected by its connectivity. Among the 11 metanodes, there are 66 possible source–target pairs. However, only 11 of them have at least one direct connection. In contrast, for paths of length 2, 50 pairs have connectivity (paths types that start on the source node type and end on the target node type, see *Figure 1C*). At length 3, all 66 pairs are connected. At length 4, the source–target pair with the fewest types of connectivity (Side Effect to Symptom) has 13 metapaths, while the pair with the most connectivity types (Gene to Gene) has 3542 pairs. This high level of connectivity across a diversity of biomedical entities forms the foundation for automated translation of knowledge into biomedical insight.

Hetionet v1.0 is accessible via a Neo4j Browser at https://neo4j.het.io. This public Neo4j instance provides users an installation-free method to query and visualize the network. The Browser contains a tutorial guide as well as guides with the details of each Project Rephetio prediction. Hetionet v1.0 is also available for download in JSON, Neo4j, and TSV formats (*Himmelstein, 2017a*). The JSON and Neo4j database formats include node and edge properties — such as URLs, source and license information, and confidence scores — and are thus recommended.

## Systematic mechanisms of efficacy

One aim of Project Rephetio was to systematically evaluate how drugs exert their therapeutic potential. To address this question, we compiled a gold standard of 755 disease-modifying indications, which form the *Compound–treats–Disease* edges in Hetionet v1.0. Next, we identified types of paths (metapaths) that occurred more frequently between treatments than non-treatments (any compound–disease pair that is not a treatment). The advantage of this approach is that metapaths naturally correspond to mechanisms of pharmacological efficacy. For example, the *Compound–binds–Gene–associates–Disease* (*CbGaD*) metapath identifies when a drug binds to a protein corresponding to a gene involved in the disease.

**Table 1.** Metanodes.
Hetionet v1.0 includes 11 node types (metanodes). For each metanode, this table shows the abbreviation, number of nodes, number of nodes without any edges, and the number of metaedges connecting the metanode.

| Metanode | Abbr | Nodes | Disconnected | Metaedges |
| --- | --- | --- | --- | --- |
| Anatomy | A | 402 | 2 | 4 |
| Biological process | BP | 11,381 | 0 | 1 |
| Cellular component | CC | 1391 | 0 | 1 |
| Compound | C | 1552 | 14 | 8 |
| Disease | D | 137 | 1 | 8 |
| Gene | G | 20,945 | 1800 | 16 |
| Molecular function | MF | 2884 | 0 | 1 |
| Pathway | PW | 1822 | 0 | 1 |
| Pharmacologic class | PC | 345 | 0 | 1 |
| Side effect | SE | 5734 | 33 | 1 |
| Symptom | S | 438 | 23 | 1 |

DOI: https://doi.org/10.7554/eLife.26726.004

**Table 2.** Metaedges.

Hetionet v1.0 contains 24 edge types (metaedges). For each metaedge, the table reports the abbreviation, the number of edges, the number of source nodes connected by the edges, and the number of target nodes connected by the edges. Note that all metaedges besides Gene→regulates→Gene are undirected.

| Metaedge | Abbr | Edges | Sources | Targets |
| --- | --- | --- | --- | --- |
| Anatomy–downregulates–Gene | AdG | 102,240 | 36 | 15,097 |
| Anatomy–expresses–Gene | AeG | 526,407 | 241 | 18,094 |
| Anatomy–upregulates–Gene | AuG | 97,848 | 36 | 15,929 |
| Compound–binds–Gene | CbG | 11,571 | 1389 | 1689 |
| Compound–causes–Side Effect | CcSE | 138,944 | 1071 | 5701 |
| Compound–downregulates–Gene | CdG | 21,102 | 734 | 2880 |
| Compound–palliates–Disease | CpD | 390 | 221 | 50 |
| Compound–resembles–Compound | CrC | 6486 | 1042 | 1054 |
| Compound–treats–Disease | CtD | 755 | 387 | 77 |
| Compound–upregulates–Gene | CuG | 18,756 | 703 | 3247 |
| Disease–associates–Gene | DaG | 12,623 | 134 | 5392 |
| Disease–downregulates–Gene | DdG | 7623 | 44 | 5745 |
| Disease–localizes–Anatomy | DlA | 3602 | 133 | 398 |
| Disease–presents–Symptom | DpS | 3357 | 133 | 415 |
| Disease–resembles–Disease | DrD | 543 | 112 | 106 |
| Disease–upregulates–Gene | DuG | 7731 | 44 | 5630 |
| Gene–covaries–Gene | GcG | 61,690 | 9043 | 9532 |
| Gene–interacts–Gene | GiG | 147,164 | 9526 | 14,084 |
| Gene–participates–Biological Process | GpBP | 559,504 | 14,772 | 11,381 |
| Gene–participates–Cellular Component | GpCC | 73,566 | 10,580 | 1391 |
| Gene–participates–Molecular Function | GpMF | 97,222 | 13,063 | 2884 |
| Gene–participates–Pathway | GpPW | 84,372 | 8979 | 1822 |
| Gene→regulates→Gene | Gr > G | 265,672 | 4634 | 7048 |
| Pharmacologic Class–includes–Compound | PCiC | 1029 | 345 | 724 |

DOI: https://doi.org/10.7554/eLife.26726.005

We evaluated all 1206 metapaths that traverse from compound to disease and have length of 2–4 (*Figure 2A*). To control for the different degrees of nodes, we used the degree-weighted path count (*DWPC*, see Materials and methods) — which downweights paths going through highly connected nodes (*Himmelstein and Baranzini, 2015a*) — to assess path prevalence. In addition, we compared the performance of each metapath to a baseline computed from permuted networks. Hetnet permutation preserves node degree while eliminating edge specificity, allowing us to isolate the portion of unpermuted metapath performance resulting from actual network paths. We refer to the permutation-adjusted performance measure as Δ AUROC. A positive Δ AUROC indicates that paths of the given type tended to occur more frequently between treatments than non-treatments, after accounting for different levels of connectivity (node degrees) in the hetnet. In general terms, Δ AUROC assesses whether paths of a given type were informative of drug efficacy.

Overall, 709 of the 1206 metapaths exhibited a statistically significant Δ AUROC at a false discovery rate cutoff of 5%. These 709 metapaths included all 24 metaedges, suggesting that each type of relationship we integrated provided at least some therapeutic utility. However, not all metaedges were equally present in significant metapaths: 259 significant metapaths included a *Compound–binds–Gene* metaedge, whereas only four included a *Gene–participates–Cellular Component*

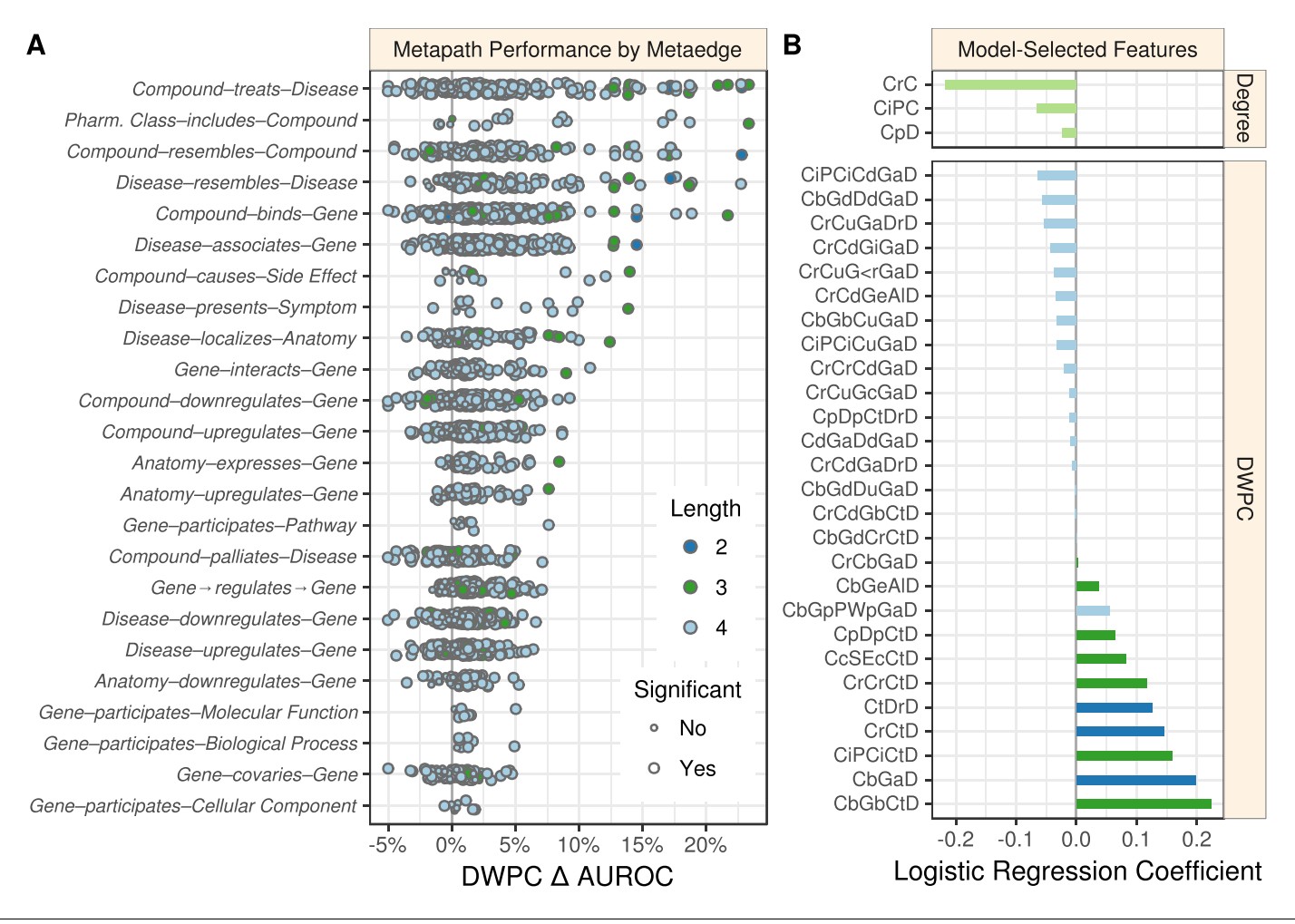

**Figure 2.** Performance by type and model coefficients. (**A**) The performance of the DWPCs for 1206 metapaths, organized by their composing metaedges. The larger dots represent metapaths that were significantly affected by permutation (false discovery rate < 5%). Metaedges are ordered by their best performing metapath. Since a metapath's performance is limited by its least informative metaedge, the best performing metapath for a metaedge provides a lower bound on the pharmacologic utility of a given domain of information. (**B**) Barplot of the model coefficients. Features were standardized prior to model fitting to make the coefficients comparable (*Himmelstein and Lizee, 2016a*).

DOI: https://doi.org/10.7554/eLife.26726.006

metaedge. *Table 3* lists the predictiveness of several metapaths of interest. Refer to the Discussion for our interpretation of these findings.

## Predictions of drug efficacy

We implemented a machine learning approach to translate the network connectivity between a compound and a disease into a probability of treatment (*Himmelstein, 2016k*; *Himmelstein, 2017b*). The approach relies on the 755 known treatments as positives and 29,044 non-treatments as negatives to train a logistic regression model. Note that 179,369 non-treatments were omitted as negative training observations because they had a prior probability of treatment equal to zero (see Materials and methods). The features consisted of a prior probability of treatment, node degrees for 14 metaedges, and DWPCs for 123 metapaths that were well suited for modeling. A cross-validated elastic net was used to minimize overfitting, yielding a model with 31 features (*Figure 2B*). The DWPC features with negative coefficients appear to be included as node-degree-capturing covariates, i.e. they reflect the general connectivity of the compound and disease rather than specific paths between them. However, the 11 DWPC features with non-negligible positive coefficients

**Table 3.** The predictiveness of select metapaths.

A small selection of interesting or influential metapaths is provided (complete table online). Len. refers to number of metaedges composing the metapath. Δ AUROC and −log10(p) assess the performance of a metapath's DWPC in discriminating treatments from non-treatments (in the all-features stage as described in Materials and methods). p assesses whether permutation affected AUROC. For reference, p=0.05 corresponds to −log10(p) = 1.30. Note that several metapaths shown here provided little evidence that Δ AUROC ≠ 0 underscoring their poor ability to predict whether a compound treated a disease. Coef. reports a metapath's logistic regression coefficient as seen in *Figure 2B*. Metapaths removed in feature selection have missing coefficients, whereas metapaths given zero-weight by the elastic net have coef. = 0.0.

| Abbrev. | Len. | Δ auroc | $-\log_{10}(P)$ | Coef. | Metapath |
|---|---|---|---|---|---|
| CbGaD | 2 | 14.5% | 6.2 | 0.20 | Compound–binds–Gene–associates–Disease |
| CdGuD | 2 | 1.7% | 4.5 | | Compound–downregulates–Gene–upregulates–Disease |
| CrCtD | 2 | 22.8% | 6.9 | 0.15 | Compound–resembles–Compound–treats–Disease |
| CtDrD | 2 | 17.2% | 5.8 | 0.13 | Compound–treats–Disease–resembles–Disease |
| CuGdD | 2 | 1.1% | 2.6 | | Compound–upregulates–Gene–downregulates–Disease |
| CbGbCtD | 3 | 21.7% | 6.5 | 0.22 | Compound–binds–Gene–binds–Compound–treats–Disease |
| CbGeAlD | 3 | 8.4% | 5.2 | 0.04 | Compound–binds–Gene–expresses–Anatomy–localizes–Disease |
| CbGiGaD | 3 | 9.0% | 4.4 | 0.00 | Compound–binds–Gene–interacts–Gene–associates–Disease |
| CcSEcCtD | 3 | 14.0% | 6.8 | 0.08 | Compound–causes–Side Effect–causes–Compound–treats–Disease |
| CdGdCtD | 3 | 3.8% | 4.6 | 0.00 | Compound–downregulates–Gene–downregulates–Compound–treats–Disease |
| CdGuCtD | 3 | −2.1% | 2.4 | | Compound–downregulates–Gene–upregulates–Compound–treats–Disease |
| CiPCiCtD | 3 | 23.3% | 7.5 | 0.16 | Compound–includes–Pharmacologic Class–includes–Compound–treats–Disease |
| CpDpCtD | 3 | 4.3% | 3.9 | 0.06 | Compound–palliates–Disease–palliates–Compound–treats–Disease |
| CrCrCtD | 3 | 17.0% | 5.0 | 0.12 | Compound–resembles–Compound–resembles–Compound–treats–Disease |
| CrCbGaD | 3 | 8.2% | 6.1 | 0.002 | Compound–resembles–Compound–binds–Gene–associates–Disease |
| CtDdGdD | 3 | 4.2% | 3.9 | | Compound–treats–Disease–downregulates–Gene–downregulates–Disease |
| CtDdGuD | 3 | 0.5% | 1.0 | | Compound–treats–Disease–downregulates–Gene–upregulates–Disease |
| CtDlAlD | 3 | 12.4% | 6.0 | | Compound–treats–Disease–localizes–Anatomy–localizes–Disease |
| CtDpSpD | 3 | 13.9% | 6.1 | | Compound–treats–Disease–presents–Symptom–presents–Disease |
| CtDuGdD | 3 | 0.7% | 1.3 | | Compound–treats–Disease–upregulates–Gene–downregulates–Disease |
| CtDuGuD | 3 | 1.1% | 1.4 | | Compound–treats–Disease–upregulates–Gene–upregulates–Disease |
| CuGdCtD | 3 | −1.6% | 2.9 | | Compound–upregulates–Gene–downregulates–Compound–treats–Disease |
| CuGuCtD | 3 | 4.4% | 3.5 | 0.00 | Compound–upregulates–Gene–upregulates–Compound–treats–Disease |
| CbGiGiGaD | 4 | 7.0% | 5.1 | 0.00 | Compound–binds–Gene–interacts–Gene–interacts–Gene–associates–Disease |
| CbGpBPpGaD | 4 | 4.9% | 3.8 | 0.00 | Compound–binds–Gene–participates–Biological Process–participates–Gene–associates–Disease |
| CbGpPWpGaD | 4 | 7.6% | 7.9 | 0.05 | Compound–binds–Gene–participates–Pathway–participates–Gene–associates–Disease |

DOI: https://doi.org/10.7554/eLife.26726.007

represent the most salient types of connectivity for systematically modeling drug efficacy. See the metapaths with positive coefficients in *Table 3* for unabbreviated names. As an example, the *CcSEcCtD* feature assesses whether the compound causes the same side effects as compounds that treat the disease. Alternatively, the *CbGeAlD* feature assesses whether the compound binds to genes that are expressed in the anatomies affected by the disease.

We applied this model to predict the probability of treatment between each of 1538 connected compounds and each of 136 connected diseases, resulting in predictions for 209,168 compound–disease pairs (*Himmelstein et al., 2016a*), available at http://het.io/repurpose/. The 755 known disease-modifying indications were highly ranked (AUROC = 97.4%, *Figure 3*). The predictions also successfully prioritized two external validation sets: novel indications from DrugCentral (AUROC = 85.5%) and novel indications in clinical trial (AUROC = 70.0%). Together, these findings indicate that Project Rephetio has the ability to recognize efficacious compound–disease pairs.

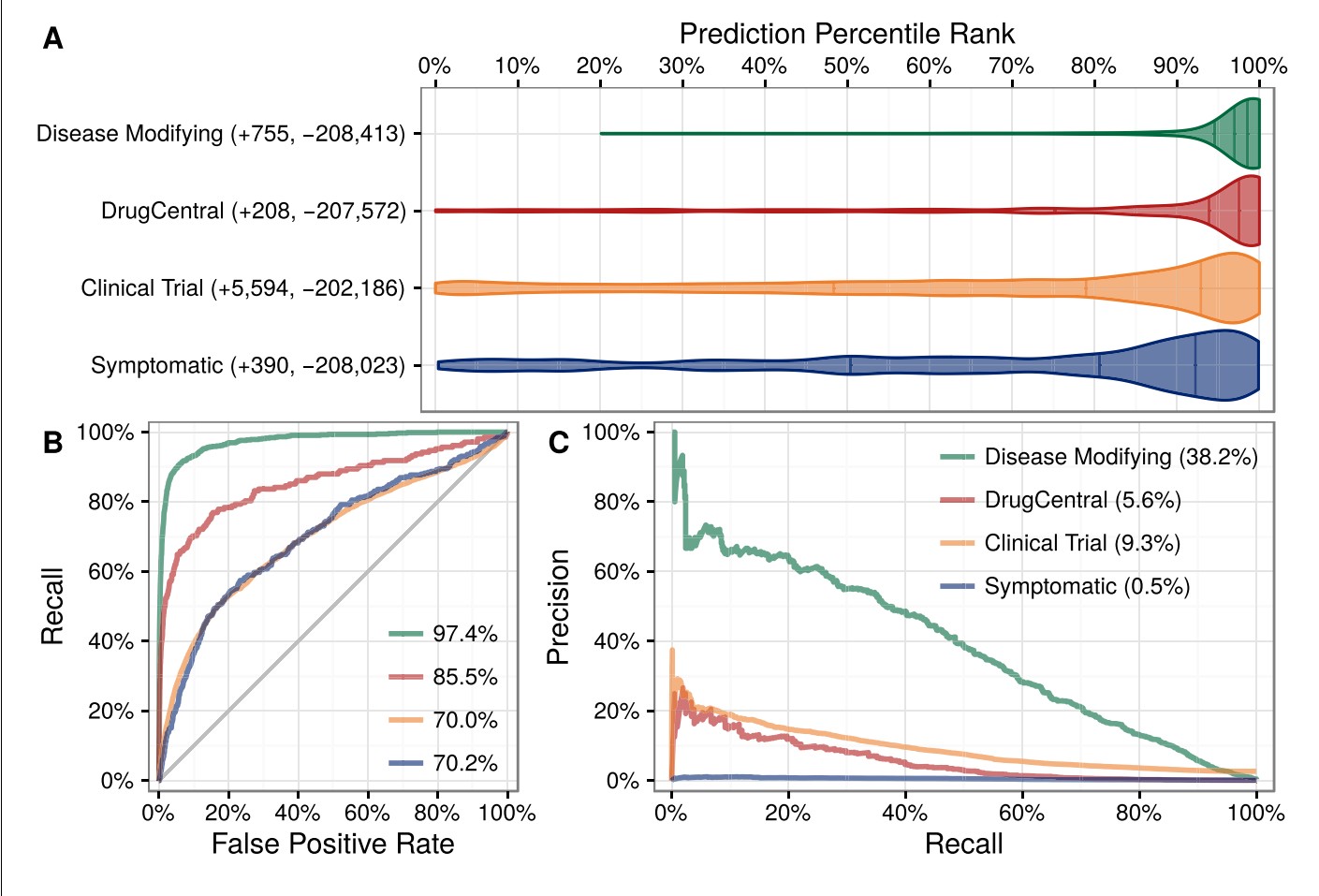

**Figure 3.** Predictions performance on four indication sets. We assess how well our predictions prioritize four sets of indications. (**A**) The y-axis labels denote the number of indications (+) and non-indications (−) composing each set. Violin plots with quartile lines show the distribution of indications when compound–disease pairs are ordered by their prediction. In all four cases, the actual indications were ranked highly by our predictions. (**B**) ROC Curves with AUROCs in the legend. (**C**) Precision–Recall Curves with AUPRCs in the legend.

DOI: https://doi.org/10.7554/eLife.26726.008

Predictions were scaled to the overall prevalence of treatments (0.36%). Hence a compound–disease pair that received a prediction of 1% represents a twofold enrichment over the null probability. Of the 3980 predictions with a probability exceeding 1%, 586 corresponded to known disease-modifying indications, leaving 3394 repurposing candidates. For a given compound or disease, we provide the percentile rank of each prediction. Therefore, users can assess whether a given prediction is a top prediction for the compound or disease. In addition, our table-based prediction browser links to a custom guide for each prediction, which displays in the Neo4j Hetionet Browser. Each guide includes a query to display the top paths supporting the prediction and lists clinical trials investigating the indication.

## Nicotine dependence case study

There are currently two FDA-approved medications for smoking cessation (varenicline and bupropion) that are not nicotine replacement therapies. PharmacotherapyDB v1.0 lists varenicline as a disease-modifying indication and nicotine itself as a symptomatic indication for nicotine dependence, but is missing bupropion. Bupropion was first approved for depression in 1985. Owing to the serendipitous observation that it decreased smoking in depressed patients taking this drug, Bupropion was approved for smoking cessation in 1997 (*Harmey et al., 2012*). Therefore, we looked whether Project Rephetio could have predicted this repurposing. Bupropion was the ninth best prediction for

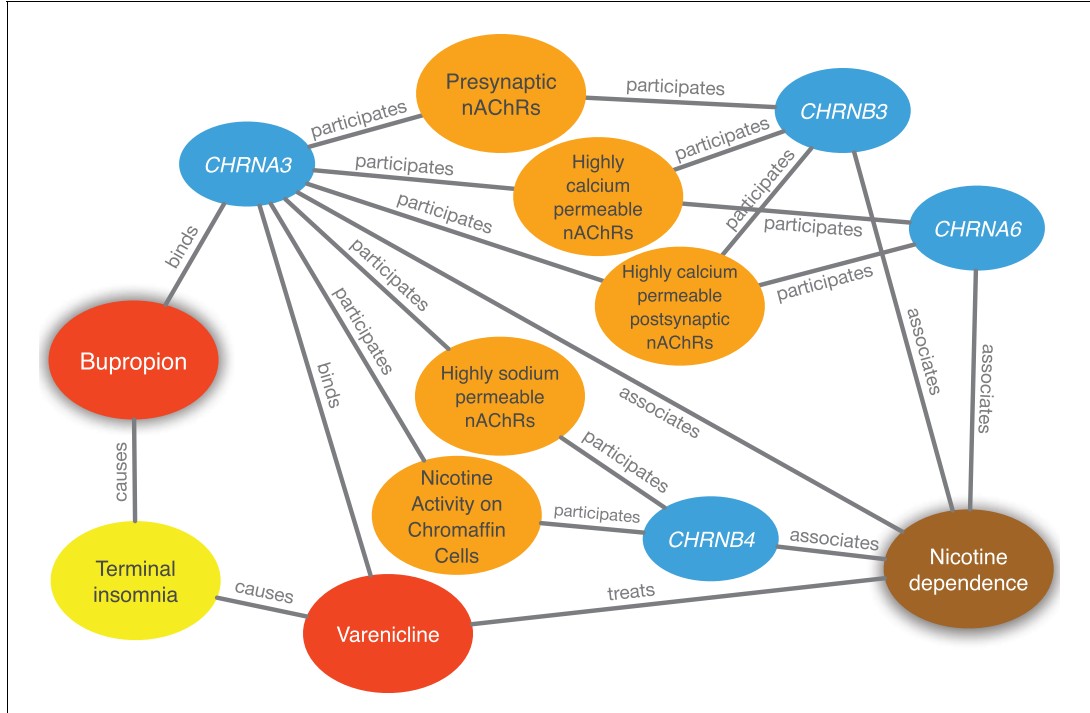

**Figure 4.** Evidence supporting the repurposing of bupropion for smoking cessation. This figure shows the 10 most supportive paths (out of 365 total) for treating nicotine dependence with bupropion, as available in this prediction's Neo4j Browser guide. Our method detected that bupropion targets the CHRNA3 gene, which is also targeted by the known-treatment varenicline (*Mihalak et al., 2006*). Furthermore, CHRNA3 is associated with nicotine dependence (*Thorgeirsson et al., 2008*) and participates in several pathways that contain other nicotinic-acetylcholine-receptor (nAChR) genes associated with nicotine dependence. Finally, bupropion causes terminal insomnia (*Boshier et al., 2003*) as does varenicline (*Hays et al., 2008*), which could indicate an underlying common mechanism of action.
DOI: https://doi.org/10.7554/eLife.26726.009

nicotine dependence (99.5th percentile) with a probability 2.50-fold greater than the null. *Figure 4* shows the top paths supporting the repurposing of bupropion.

Atop the nicotine dependence predictions were nicotine (10.97-fold over null), cytisine (10.58-fold), and galantamine (9.50-fold). Cytisine is widely used in Eastern Europe for smoking cessation due to its availability at a fraction of the cost of other pharmaceutical options (*Cahill et al., 2016*). In the last half decade, large-scale clinical trials have confirmed cytisine's efficacy (*West et al., 2011*; *Walker et al., 2014*). Galantamine, an approved Alzheimer's treatment, is currently in Phase 2 trial for smoking cessation and is showing promising results (*Ashare et al., 2016*). In summary, nicotine dependence illustrates Project Rephetio's ability to predict efficacious treatments and prioritize historic and contemporary repurposing opportunities.

## Epilepsy case study

Several factors make epilepsy an interesting disease for evaluating repurposing predictions (*Khankhanian and Himmelstein, 2016*). Antiepileptic drugs work by increasing the seizure threshold — the amount of electric stimulation that is required to induce seizure. The effect of a drug on the seizure threshold can be cheaply and reliably tested in rodent models. As a result, the viability of most approved drugs in treating epilepsy is known.

We focused our evaluation on the top 100 scoring compounds — referred to as the epilepsy predictions in this section — after discarding a single combination drug. We classified each compound as anti-ictogenic (seizure suppressing), unknown (no established effect on the seizure threshold), or ictogenic (seizure generating) according to medical literature (*Khankhanian and Himmelstein, 2016*). Of the top 100 epilepsy predictions, 77 were anti-ictogenic, eight were unknown, and 15

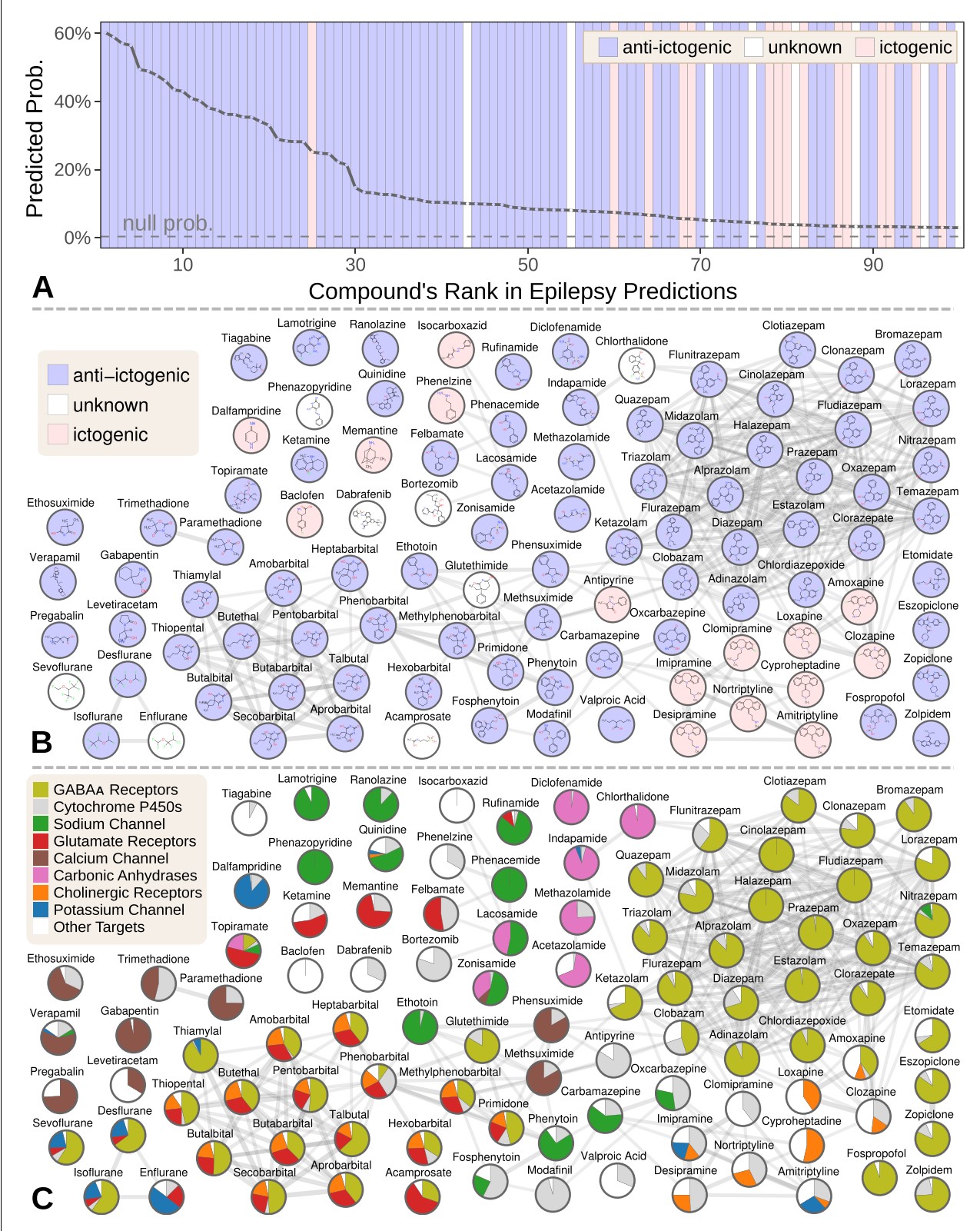

**Figure 5.** Top 100 epilepsy predictions. (**A**) Compounds — ranked from 1 to 100 by their predicted probability of treating epilepsy — are colored by their effect on seizures (*Khankhanian and Himmelstein, 2016*). The highest predictions are almost exclusively anti-ictogenic. Further down the prediction list, the prevalence of drugs with an ictogenic (contraindication) or unknown (novel repurposing candidate) effect on epilepsy increases. All compounds shown received probabilities far exceeding the null probability of treatment (0.36%). (**B**) A chemical similarity network of the epilepsy

*Figure 5 continued on next page*

*Figure 5 continued*

predictions, with each compound's 2D structure (*Himmelstein et al., 2017a*). Edges are Compound–resembles–Compound relationships from Hetionet v1.0. Nodes are colored by their effect on seizures. (C) The relative contribution of important drug targets to each epilepsy prediction (*Himmelstein et al., 2017a*). Specifically, pie charts show how the eight most-supportive drug targets across all 100 epilepsy predictions contribute to individual predictions. Other Targets represents the aggregate contribution of all targets not listed. The network layout is identical to B.
DOI: https://doi.org/10.7554/eLife.26726.010

were ictogenic (*Figure 5A*). Notably, the predictions contained 23 of the 25 disease-modifying antie-pileptics in PharamcotherapyDB v1.0.

Many of the 77 anti-ictogenic compounds were not first-line antiepileptic drugs. Instead, they were used as ancillary drugs in the treatment of status epilepticus. For example, we predicted four halogenated ethers, two of which (isoflurane and desflurane) are used clinically to treat life-threatening seizures that persist despite treatment (*Mirsattari et al., 2004*). As inhaled anesthetics, these compounds are not appropriate as daily epilepsy medications, but are feasible for refractory status epilepticus where patients are intubated.

Given this high precision (77%), the eight compounds of unknown effect are promising repurposing candidates. For example, acamprosate — whose top prediction was epilepsy — is a taurine analog that promotes alcohol abstinence. Support for this repurposing arose from acamprosate's inhibition of the glutamate receptor and positive modulation of the GABAA receptor (*Figure 5C*). If effective against epilepsy, acamprosate could serve a dual benefit for recovering alcoholics who experience seizures from alcohol withdrawal.

While certain classes of compounds were highly represented in our epilepsy predictions, such benzodiazepines and barbiturates, there was also considerable diversity (*Khankhanian and Himmelstein, 2016*). The 100 predicted compounds encompassed 26 third-level ATC codes (*Knaus, 2016*), such as antiarrhythmics (quinidine, classified as anti-ictogenic) and urologicals (phenazopyridine, classified as unknown). Furthermore, 25 of the compounds were chemically distinct, i.e. they did not resemble any of the other epilepsy predictions (*Figure 5B*).

Next, we investigated which components of Hetionet contributed to the epilepsy predictions (*Khankhanian and Himmelstein, 2016*). In total, 392,956 paths of 12 types supported the predictions. Using several different methods for grouping paths, we were able to quantify the aggregate biological evidence. Our algorithm primarily drew on two aspects of epilepsy: its known treatments (76% of the total support) and its genetic associations (22% of support). In contrast, our algorithm drew heavily on several aspects of the predicted compounds: their targeted genes (44%), their chemically similar compounds (30%), their pharmacologic classes, their palliative indications (5%), and their side effects (4%).

Specifically, 266,192 supporting paths originated with a *Compound–binds–Gene* relationship. Aggregating support by these genes shows the extent that 121 different drug targets contributed to the predictions (*Khankhanian and Himmelstein, 2016*). In order of importance, the predictions targeted GABAA receptors (15.3% of total support), cytochrome P450 enzymes (5.6%), the sodium channel (4.6%), glutamate receptors (3.8%), the calcium channel (2.7%), carbonic anhydrases (2.5%), cholinergic receptors (2.1%), and the potassium channel (1.4%). Besides cytochrome P450, which primarily influences pharmacokinetics (*Johannessen and Landmark, 2010*), our method detected and leveraged bonafide anti-ictogenic mechanisms (*Rogawski and Löscher, 2004*). *Figure 5C* shows drug target contributions per compound and illustrates the considerable mechanistic diversity among the predictions.

Also notable are the 15 ictogenic compounds in our top 100 predictions. Nine of the ictogenic compounds share a tricyclic structure (*Figure 5B*), five of which are tricyclic antidepressants. While the ictogenic mechanisms of these antidepressants are still unclear (*Johannessen Landmark et al., 2016*), *Figure 5C* suggests their anticholinergic effects may be responsible (*Himmelstein, 2017d*), in accordance with previous theories (*Dailey and Naritoku, 1996*).

We also ranked the contribution of the 1137 side effects that supported the epilepsy predictions through 117,720 *CcSEcCtD* paths. The top five side effects — ataxia (0.069% of total support), nystagmus (0.049%), diplopia (0.045%), somnolence (0.044%), and vomiting (0.043%) — reflect established adverse effects of antiepileptic drugs (*Zadikoff et al., 2007*; *Wu and Thijs, 2015*; *ROFF HILTONHilton et al., 2004*; *Placidi et al., 2000*; *Jahromi et al., 2011*). In summary, our

method simultaneously identified the hallmark side effects of antiepileptic drugs while incorporating this knowledge to prioritize 1538 compounds for anti-ictogenic activity.

## Discussion

We created Hetionet v1.0 by integrating 29 resources into a single data structure — the hetnet. Consisting of 11 types of nodes and 24 types of relationships, Hetionet v1.0 brings more types of information together than previous leading-studies in biological data integration (*Gligorijević and Pržulj, 2015*). Moreover, we strove to create a reusable, extensible, and property-rich network. While all the resources we include are publicly available, their integration was a time-intensive undertaking and required careful consideration of legal barriers to data reuse. Hetionet allows researchers to begin answering integrative questions without having to first spend months processing data.

Our public Neo4j instance allows users to immediately interact with Hetionet. Through the Cypher language, users can perform highly specialized graph queries with only a few lines of code. Queries can be executed in the web browser or programmatically from a language with a Neo4j driver. For users that are unfamiliar with Cypher, we include several example queries in a Browser guide. In contrast to traditional REST APIs, our public Neo4j instance provides users with maximal flexibility to construct custom queries by exposing the underlying database.

As data has grown more plentiful and diverse, so has the applicability of hetnets. Unfortunately, network science has been naturally fragmented by discipline resulting in relatively slow progress in integrating heterogeneous data. A 2014 analysis identified 78 studies using multilayer networks — a superset of hetnets (heterogeneous information networks) with the potential for additional dimensions, such as time. However, the studies relied on 26 different terms, 9 of which had multiple definitions (*Kivela et al., 2014*; *Himmelstein et al., 2015b*). Nonetheless, core infrastructure and algorithms for hetnets are emerging. Compared to the existing mathematical frameworks for multilayer networks that must deal with layers other than type (such as the aspect of time) (*Kivela et al., 2014*), the primary obligation of hetnet algorithms is to be type aware. One goal of our project has been to unite hetnet research across disciplines. We approached this goal by making Project Rephetio entirely available online and inviting community feedback throughout the process (*Himmelstein et al., 2015c*).

Integrating every resource into a single interconnected data structure allowed us to assess systematic mechanisms of drug efficacy. Using the max performing metapath to assess the pharmacological utility of a metaedge (*Figure 2A*), we can divide our relationships into tiers of informativeness. The top tier consists of the types of information traditionally considered by pharmacology: *Compound–treats–Disease*, *Pharmacologic Class–includes–Compound*, *Compound–resembles–Compound*, *Disease–resembles–Disease*, and *Compound–binds–Gene*. The upper-middle tier consists of types of information that have been the focus of substantial medical study, but have only recently started to play a bigger role in drug development, namely the metaedges *Disease–associates–Gene*, *Compound–causes–Side Effect*, *Disease–presents–Symptom*, *Disease–localizes–Anatomy*, and *Gene–interacts–Gene*.

The lower-middle tier contains the transcriptomics metaedges such as *Compound–downregulates–Gene*, *Anatomy–expresses–Gene*, *Gene→regulates→Gene*, and *Disease–downregulates–Gene*. Much excitement surrounds these resources due to their high-throughput and genome-wide scope, which offers a route to drug discovery that is less biased by existing knowledge. However, our findings suggest that these resources are only moderately informative of drug efficacy. Other lower-middle tier metaedges were the product of time-intensive biological experimentation, such as *Gene–participates–Pathway*, *Gene–participates–Molecular Function*, and *Gene–participates–Biological Process*. Unlike the top tier resources, this knowledge has historically been pursued for basic science rather than primarily medical applications. The weak yet appreciable performance of the *Gene–covaries–Gene* suggests the synergy between the fields of evolutionary genomics and disease biology. The lower tier included the *Gene–participates–Cellular Component* metaedge, which may reflect that the relevance of cellular location to pharmacology is highly case dependent and not amenable to systematic profiling.

The performance of specific metapaths (*Table 3*) provides further insight. For example, significant emphasis has been put on the use of transcriptional data for drug repurposing (*Iorio et al., 2013*). One common approach has been to identify compounds with opposing transcriptional signatures to

a disease (*Qu and Rajpal, 2012*; *Sirota et al., 2011*). However, several systematic studies report underwhelming performance of this approach (*Gottlieb et al., 2011*; *Cheng et al., 2014*; *Guney et al., 2016*) — a finding supported by the low performance of the *CuGdD* and *CdGuD* metapaths in Project Rephetio. Nonetheless, other transcription-based methods showed some promise. Compounds with similar transcriptional signatures were prone to treating the same disease (*CuGuCtD* and *CdGdCtD* metapaths), while compounds with opposing transcriptional signatures were slightly averse to treating the same disease (*CuGdCtD* and *CdGuCtD* metapaths). In contrast, diseases with similar transcriptional profiles were not prone to treatment by the same compound (*CtDdGuD* and *CtDuGdD*).

By comparably assessing the informativeness of different metaedges and metapaths, Project Rephetio aims to guide future research towards promising data types and analyses. One caveat is that omics-scale experimental data will likely play a larger role in developing the next generation of pharmacotherapies. Hence, were performance reevaluated on treatments discovered in the forthcoming decades, the predictive ability of these data types may rise. Encouragingly, most data types were at least weakly informative and hence suitable for further study. Ideally, different data types would provide orthogonal information. However, our model for whether a compound treats a disease focused on 11 metapaths — a small portion of the hundreds of metapaths available. While parsimony aids interpretation, our model did not draw on the weakly-predictive high-throughput data types — which are intriguing for their novelty, scalability, and cost-effectiveness — as much as we had hypothesized.

Instead our model selected types of information traditionally considered in pharmacology. However, unlike a pharmacologist whose area of expertise may be limited to a few drug classes, our model was able to predict probabilities of treatment for all 209,168 compound–disease pairs. Furthermore, our model systematically learned the importance of each type of network connectivity. For any compound–disease pair, we now can immediately provide the top network paths supporting its therapeutic efficacy. A traditional pharmacologist may be able to produce a similar explanation, but likely not until spending substantial time researching the compound's pharmacology, the disease's pathophysiology, and the molecular relationships in between. Accordingly, we hope certain predictions will spur further research, such as trials to investigate the off-label use of acamprosate for epilepsy, which is supported by one animal model (*Farook et al., 2008*).

As demonstrated by the 15 ictogenic compounds in our top 100 epilepsy predictions, Project Rephetio's predictions can include contraindications in addition to indications. Since many of Hetionet v1.0's relationship types are general (e.g. the *Compound–binds–Gene* relationship type conflates antagonist with agonist effects), we expect some high scoring predictions to exacerbate rather than treat the disease. However, the predictions made by Hetionet v1.0 represent such substantial relative enrichment over the null that uncovering the correct directionality is a logical next step and worth undertaking. Going forward, advances in automated mining of the scientific literature could enable extraction of precise relationship types at omics scale (*Ehrenberg et al., 2016*; *Himmelstein et al., 2016b*).

Future research should focus on gleaning orthogonal information from data types that are so expansive that computational methods are the only option. Our *CuGuCtD* feature — measuring whether a compound upregulates the same genes as compounds which treat the disease — is a good example. This metapath was informative by itself ($\Delta$ AUROC = 4.4%) but was not selected by the model, despite its orthogonal origin (gene expression) to selected metapaths. Using a more extensive catalog of treatments as the gold standard would be one possible approach to increase the power of feature selection.

Integrating more types of information into Hetionet should also be a future priority. The 'network effect' phenomenon suggests that the addition of each new piece of information will enhance the value of Hetionet's existing information. We envision a future where all biological knowledge is encoded into a single hetnet. Hetionet v1.0 was an early attempt, and we hope the strong performance of Project Rephetio in repurposing drugs foreshadows the many applications that will thrive from encoding biology in hetnets.

## Materials and methods

Hetionet was built entirely from publicly available resources with the goal of integrating a broad diversity of information types of medical relevance, ranging in scale from molecular to organismal. Practical considerations such as data availability, licensing, reusability, documentation, throughput, and standardization informed our choice of resources. We abided by a simple litmus test for determining how to encode information in a hetnet: nodes represent nouns, relationships represent verbs (*Chen, 1997*; *Himmelstein et al., 2016c*).

Our method for relationship prediction creates a strong incentive to avoid redundancy, which increases the computational burden without improving performance. In a previous study to predict disease–gene associations using a hetnet of pathophysiology (*Himmelstein and Baranzini, 2015a*), we found that different types of gene sets contributed highly redundant information. Therefore, in Hetionet v1.0, we reduced the number of gene set node types from 14 to 3 by omitting several gene set collections and aggregating all pathway nodes.

### Nodes

Nodes encode entities. We extracted nodes from standard terminologies, which provide curated vocabularies to enable data integration and prevent concept duplication. The ease of mapping external vocabularies, adoption, and comprehensiveness were primary selection criteria. Hetionet v1.0 includes nodes from five ontologies — which provide hierarchy of entities for a specific domain — selected for their conformity to current best practices (*Malone et al., 2016*).

We selected 137 terms from the Disease Ontology (*Schriml et al., 2012*; *Kibbe et al., 2015*) (which we refer to as DO Slim (*Himmelstein and Li, 2015d*; *Himmelstein, 2016g*)) as our **disease** set. Our goal was to identify complex diseases that are distinct and specific enough to be clinically relevant yet general enough to be well annotated. To this end, we included diseases that have been studied by GWAS and cancer types from TopNodes_DOcancerslim (*Wu et al., 2015*). We ensured that no DO Slim disease was a subtype of another DO Slim disease. **Symptoms** were extracted from MeSH by taking the 438 descendants of *Signs and Symptoms* (*Himmelstein and Pankov, 2015a*; *Himmelstein, 2016h*).

Approved small molecule **compounds** with documented chemical structures were extracted from DrugBank version 4.2 (*Law et al., 2014*; *Himmelstein, 2015b*; *Himmelstein, 2016i*). Unapproved compounds were excluded because our focus was repurposing. In addition, unapproved compounds tend to be less studied than approved compounds making them less attractive for our approach where robust network connectivity is critical. Finally, restricting to small molecules with known documented structures enabled us to map between compound vocabularies (see Mappings).

**Side effects** were extracted from SIDER version 4.1 (*Kuhn et al., 2016*; *Himmelstein, 2015c*; *Himmelstein, 2016j*). SIDER codes side effects using UMLS identifiers (*Bodenreider, 2004*), which we also adopted. **Pharmacologic Classes** were extracted from the DrugCentral data repository (*Ursu et al., 2017*; *Himmelstein et al., 2016d*). Only pharmacologic classes corresponding to the 'Chemical/Ingredient', 'Mechanism of Action', and 'Physiologic Effect' FDA class types were included to avoid pharmacologic classes that were synonymous with indications (*Himmelstein et al., 2016d*).

Protein-coding human **genes** were extracted from Entrez Gene (*Maglott et al., 2011*; *Himmelstein et al., 2015h*; *Himmelstein, 2016l*). Anatomical structures, which we refer to as **anatomies**, were extracted from Uberon (*Mungall et al., 2012*). We selected a subset of 402 Uberon terms by excluding terms known not to exist in humans and terms that were overly broad or arcane (*Malladi et al., 2015*; *Himmelstein, 2016m*).

**Pathways** were extracted by combining human pathways from WikiPathways (*Kutmon et al., 2016*; *Pico et al., 2008*), Reactome (*Fabregat et al., 2016*), and the Pathway Interaction Database (*Schaefer et al., 2009*). The latter two resources were retrieved from Pathway Commons (RRID: SCR_002103) (*Cerami et al., 2011*), which compiles pathways from several providers. Duplicate pathways and pathways without multiple participating genes were removed (*Pico and Himmelstein, 2015*; *Himmelstein and Pico, 2016a*). **Biological processes**, **cellular components**, and **molecular functions** were extracted from the Gene Ontology (*Ashburner et al., 2000*). Only terms with 2–1000 annotated genes were included.

## Mappings

Before adding relationships, all identifiers needed to be converted into the vocabularies matching that of our nodes. Oftentimes, our node vocabularies included external mappings. For example, the Disease Ontology includes mappings to MeSH, UMLS, and the ICD, several of which we submitted during the course of this study (*Himmelstein, 2015e*). In a few cases, the only option was to map using gene symbols, a disfavored method given that it can lead to ambiguities.

When mapping external disease concepts onto DO Slim, we used transitive closure. For example, the UMLS concept for primary progressive multiple sclerosis (C0751964) was mapped to the DO Slim term for multiple sclerosis (DOID:2377).

Chemical vocabularies presented the greatest mapping challenge (*Himmelstein, 2015b*), since these are poorly standardized (*Hersey et al., 2015*). UniChem's (*Chambers et al., 2013*) Connectivity Search (*Chambers et al., 2014*) was used to map compounds, which maps by atomic connectivity (based on First InChIKey Hash Blocks (*Heller et al., 2013*)) and ignores small molecular differences.

## Edges

*Anatomy–downregulates–Gene* and *Anatomy–upregulates–Gene* edges (*Himmelstein et al., 2016f*; *Himmelstein and Bastian, 2015e*; *Himmelstein and Bastian, 2015f*) were extracted from Bgee (*Bastian et al., 2008*), which computes differentially expressed genes by anatomy in post-juvenile adult humans. *Anatomy–expresses–Gene* edges were extracted from Bgee and TISSUES (*Santos et al., 2015*; *Himmelstein and Jensen, 2015g*; *Himmelstein and Jensen, 2015h*).

*Compound–binds–Gene* edges were aggregated from BindingDB (*Chen et al., 2001*; *Gilson et al., 2016*), DrugBank (*Law et al., 2014*; *Wishart et al., 2006*), and DrugCentral (*Ursu et al., 2017*). Only binding relationships to single proteins with affinities of at least 1 µM (as determined by $K_d$, $K_i$, or $IC_{50}$) were selected from the October 2015 release of BindingDB (*Himmelstein and Gilson, 2015i*; *Himmelstein et al., 2015d*). Target, carrier, transporter, and enzyme interactions with single proteins (i.e. excluding protein groups) were extracted from DrugBank 4.2 (*Himmelstein, 2016i*; *Himmelstein and Protein, 2015j*). In addition, all mapping DrugCentral target relationships were included (*Himmelstein et al., 2016d*).

*Compound–treats–Disease* (disease-modifying indications) and *Compound–palliates–Disease* (symptomatic indications) edges are from PharmacotherapyDB as described in Intermediate resources. *Compound–causes–Side Effect* edges were obtained from SIDER 4.1 (*Kuhn et al., 2016*; *Himmelstein, 2015c*; *Himmelstein, 2016j*), which uses natural language processing to identify side effects in drug labels. *Compound–resembles–Compound* relationships (*Himmelstein, 2016i*; *Himmelstein and Chen, 2015k*; *Himmelstein et al., 2015q*) represent chemical similarity and correspond to a Dice coefficient $\geq 0.5$ (*Dice, 1945*) between extended connectivity fingerprints (*Rogers and Hahn, 2010*; *Morgan, 1965*). *Pharmacologic Class–includes–Compound* edges were extracted from DrugCentral for three FDA class types (*Ursu et al., 2017*; *Himmelstein et al., 2016d*). *Compound–downregulates–Gene* and *Compound–upregulates–Gene* relationships were computed from LINCS L1000 as described in Intermediate resources.

*Disease–associates–Gene* edges were extracted from the GWAS Catalog (*Himmelstein and Baranzini, 2016b*), DISEASES (*Himmelstein and Jensen, 2015l*; *Himmelstein and Jensen, 2016c*), DisGeNET (*Himmelstein, 2015f*; *Himmelstein and Piñero, 2016d*), and DOAF (*Himmelstein, 2015g*; *Himmelstein, 2016s*). The GWAS Catalog compiles disease–SNP associations from published GWAS (*MacArthur et al., 2017*). We aggregated overlapping loci associated with each disease and identified the mode reported gene for each high confidence locus (*Himmelstein, 2015h*; *Himmelstein et al., 2015v*). DISEASES integrates evidence of association from text mining, curated catalogs, and experimental data (*Pletscher-Frankild et al., 2015*). Associations from DISEASES with integrated scores $\geq 2$ were included after removing the contribution of DistiLD. DisGeNET integrates evidence from over 10 sources and reports a single score for each association (*Piñero et al., 2015*; *Piñero et al., 2017*). Associations with scores $\geq 0.06$ were included. DOAF mines Entrez Gene GeneRIFs (textual annotations of gene function) for disease mentions (*Xu et al., 2012*). Associations with three or more supporting GeneRIFs were included. *Disease–downregulates–Gene* and *Disease–upregulates–Gene* relationships (*Himmelstein et al., 2015a*; *Himmelstein et al., 2016j*) were computed using STARGEO as described in Intermediate resources.

*Disease–localizes–Anatomy*, *Disease–presents–Symptom*, and *Disease–resembles–Disease* edges were calculated from MEDLINE co-occurrence (*Himmelstein and Pankov, 2015a*; *Himmelstein, 2016u*). MEDLINE is a subset of 21 million PubMed articles for which designated human curators have assigned topics. When retrieving articles for a given topic (MeSH term), we activated two non-default search options as specified below: majr for selecting only articles where the topic is major and noexp for suppressing explosion (returning articles linked to MeSH subterms). We identified 4,161,769 articles with two or more disease topics; 696,252 articles with both a disease topic (majr) and an anatomy topic (noexp) (*Himmelstein, 2015i*); and 363,928 articles with both a disease topic (majr) and a symptom topic (noexp). We used a Fisher's exact test (*Fisher, 1922*) to identify pairs of terms that occurred together more than would be expected by chance in their respective corpus. We included co-occurring terms with p<0.005 in Hetionet v1.0.

*Gene→regulates→Gene* directed edges were generated from the LINCS L1000 genetic interference screens (see Intermediate resources) and indicate that knockdown or overexpression of the source gene significantly dysregulated the target gene (*Himmelstein and Chung, 2015q*; *Himmelstein et al., 2016k*). *Gene–covaries–Gene* edges represent evolutionary rate covariation ≥0.75 (*Priedigkeit et al., 2015*; *Himmelstein and Partha, 2015r*; *Himmelstein, 2016w*). *Gene–interacts–Gene* edges (*Himmelstein et al., 2015z*; *Himmelstein and Baranzini, 2016e*) represent when two genes produce physically interacting proteins. We compiled these interactions from the Human Interactome Database (*Rual et al., 2005*; *Venkatesan et al., 2009*; *Yu et al., 2011*; *Rolland et al., 2014*), the Incomplete Interactome (*Menche et al., 2015*), and our previous study (*Himmelstein and Baranzini, 2015a*). *Gene–participates–Biological Process*, *Gene–participates–Cellular Component*, and *Gene–participates–Molecular Function* edges are from Gene Ontology annotations (*Huntley et al., 2015*). As described in Intermediate resources, annotations were propagated (*Himmelstein et al., 2015g*; *Himmelstein et al., 2015f*). *Gene–participates–Pathway* edges were included from the human pathway resources described in the Nodes section (*Pico and Himmelstein, 2015*; *Himmelstein and Pico, 2016a*).

## Directionality

Whether a certain type of relationship has directionality is defined at the metaedge level. Directed metaedges are only necessary when they connect a metanode to itself and correspond to an asymmetric relationship. In the case of Hetionet v1.0, the sole directed metaedge was *Gene→regulates→Gene*. To demonstrate the implications of directionality, Hetionet v1.0 contains two relationships between the genes *HADH* and *STAT1*: *HADH–interacts–STAT1* and *HADH→regulates→STAT1*. Both edges can be represented in the inverse orientation: *STAT1–interacts–HADH* and *STAT1←regulates←HADH*. However due to directed nature of the *regulates* relationship, *STAT1→regulates→HADH* is a distinct edge, which does not exist in the network. Similarly, *HADH–associates–obesity* and *obesity–associates–HADH* are inverse orientations of the same underlying undirected relationship. Accordingly, the following path exists in the network: *obesity–associates–HADH→regulates→STAT1*, which can also be inverted to *STAT1←regulates←HADH–associates–obesity*.

## Intermediate resources

In the process of creating Hetionet, we produced several datasets with broad applicability that extended beyond Project Rephetio. These resources are referred to as intermediate resources and described below.

## Transcriptional signatures of disease using STARGEO

STARGEO is a nascent platform for annotating and meta-analyzing differential gene expression experiments (*Hadley et al., 2017*). The STAR acronym stands for Search-Tag-Analyze Resources, while GEO refers to the Gene Expression Omnibus (*Edgar et al., 2002*; *Barrett et al., 20122013*). STARGEO is a layer on top of GEO that crowdsources sample annotation and automates meta-analysis.

Using STARGEO, we computed differentially expressed genes between healthy and diseased samples for 49 diseases (*Himmelstein et al., 2015a*; *Himmelstein et al., 2016j*). First, we and others created case/control tags for 66 diseases. After combing through GEO series and tagging samples,

49 diseases had sufficient data for case-control meta-analysis: multiple series with at least three cases and three controls. For each disease, we performed a random effects meta-analysis on each gene to combine $\log_2$ fold-change across series. These analyses incorporated 27,019 unique samples from 460 series on 107 platforms.

Differentially expressed genes (false discovery rate $\leq$0.05) were identified for each disease. The median number of upregulated genes per disease was 351 and the median number of downregulated genes was 340. Endogenous depression was the only of the 49 diseases without any significantly dysregulated genes.

## Transcriptional signatures of perturbation from LINCS L1000

LINCS L1000 profiled the transcriptional response to small molecule and genetic interference perturbations. To increase throughput, expression was only measured for 978 genes, which were selected for their ability to impute expression of the remaining genes. A single perturbation was often assayed under a variety of conditions including cell types, dosages, timepoints, and concentrations. Each condition generates a single signature of dysregulation z-scores. We further processed these signatures to fit into our approach (*Himmelstein et al., 2016m*; *Himmelstein et al., 2016n*).

First, we computed consensus signatures — which meta-analyze multiple signatures to condense them into one — for DrugBank small molecules, Entrez genes, and all L1000 perturbations (*Himmelstein and Chung, 2015q*; *Himmelstein et al., 2016k*). First, we discarded non-gold (non-replicating or indistinct) signatures. Then, we meta-analyzed z-scores using Stouffer's method. Each signature was weighted by its average Spearman's correlation to other signatures, with a 0.05 minimum, to de-emphasize discordant signatures. Our signatures include the 978 measured genes and the 6489 imputed genes from the 'best inferred gene subset'. To identify significantly dysregulated genes, we selected genes using a Bonferroni cutoff of p=0.05 and limited the number of imputed genes to 1000.

The consensus signatures for genetic perturbations allowed us to assess various characteristics of the L1000 dataset. First, we looked at whether genetic interference dysregulated its target gene in the expected direction (*Himmelstein, 2016c*). Looking at measured z-scores for target genes, we found that the knockdown perturbations were highly reliable, while the overexpression perturbations were only moderately reliable with 36% of overexpression perturbations downregulating their target. However, imputed z-scores for target genes barely exceeded chance at responding in the expected direction to interference. Hence, we concluded that the imputation quality of LINCS L1000 is poor. However, when restricting to significantly dysregulated targets, 22 out of 29 imputed genes responded in the expected direction. This provides some evidence that the directional fidelity of imputation is higher for significantly dysregulated genes. Finally, we found that the transcriptional signatures of knocking down and overexpressing the same gene were positively correlated 65% of the time, suggesting the presence of a general stress response (*Himmelstein et al., 2016o*).

Based on these findings, we performed additional filtering of significantly dysregulated genes when building Hetionet v1.0. *Compound–down/up-regulates–Gene* relationships were restricted to the 125 most significant per compound-direction-status combination (status refers to measured versus imputed). For genetic interference perturbations, we restricted to the 50 most significant genes per gene-direction-status combination and merged the remaining edges into a single *Gene→regulates→Gene* relationship type containing both knockdown and overexpression perturbations.

## PharmacotherapyDB: physician curated indications

We created PharmacotherapyDB, an open catalog of drug therapies for disease (*Himmelstein, 2016a*; *Himmelstein et al., 2016p*; *Himmelstein et al., 2016q*). Version 1.0 contains 755 disease-modifying therapies and 390 symptomatic therapies between 97 diseases and 601 compounds.

This resource was motivated by the need for a gold standard of medical indications to train and evaluate our approach. Initially, we identified four existing indication catalogs (*Himmelstein et al., 2015e*): MEDI-HPS which mined indications from RxNorm, SIDER 2, MedlinePlus, and Wikipedia (*Wei et al., 2013*); LabeledIn which extracted indications from drug labels via human curation (*Khare et al., 2014*; *Khare et al., 2015*; *Himmelstein and Khare, 2015s*); EHRLink which identified

medication–problem pairs that clinicians linked together in electronic health records (*McCoy et al., 2012*; *Himmelstein, 2015j*); and indications from PREDICT, which were compiled from UMLS relationships, drugs.com, and drug labels (*Gottlieb et al., 2011*). After mapping to DO Slim and Drug-Bank Slim, the four resources contained 1388 distinct indications.

However, we noticed that many indications were palliative and hence problematic as a gold standard of pharmacotherapy for our in silico approach. Therefore, we recruited two practicing physicians to curate the 1388 preliminary indications (*Himmelstein et al., 2015j*). After a pilot on 50 indications, we defined three classifications: *disease modifying* meaning a drug that therapeutically changes the underlying or downstream biology of the disease; *symptomatic* meaning a drug that treats a significant symptom of the disease; and *non-indication* meaning a drug that neither therapeutically changes the underlying or downstream biology nor treats a significant symptom of the disease. Both curators independently classified all 1388 indications.

The two curators disagreed on 444 calls (Cohen's κ = 49.9%). We then recruited a third practicing physician, who reviewed all 1388 calls and created a detailed explanation of his methodology (*Himmelstein et al., 2015j*). We proceeded with the third curator's calls as the consensus curation. The first two curators did have reservations with classifying steroids as disease modifying for autoimmune diseases. We ultimately considered that these indications met our definition of disease modifying, which is based on a pathophysiological rather than clinical standard. Accordingly, therapies we consider disease modifying may not be used to alter long-term disease course in the modern clinic due to a poor risk–benefit ratio.

## User-friendly gene ontology annotations

We created a browser (http://git.dhimmel.com/gene-ontology/) to provide straightforward access to Gene Ontology annotations (*Himmelstein et al., 2015g*; *Himmelstein et al., 2015f*). Our service provides annotations between Gene Ontology terms and Entrez Genes. The user chooses propagated/direct annotation and all/experimental evidence. Annotations are currently available for 37 species and downloadable as user-friendly TSV files.

## Data copyright and licensing

We committed to openly releasing our data and analyses from the origin of the project (*Spaulding et al., 2015*). Our goals were to contribute to the advancement of science (*Hrynaszkiewicz, 2011*; *Molloy, 2011*), maximize our impact (*McKiernan et al., 2016*; *Piwowar and Vision, 2013*), and enable reproducibility (*Stodden et al., 2016*; *Stodden and Miguez, 2014*; *Baggerly, 2010*). These objectives required publicly distributing and openly licensing Hetionet and Project Rephetio data and analyses (*Hrynaszkiewicz and Cockerill, 2012*; *Hagedorn et al., 2011*).

Since we integrated only public resources, which were overwhelmingly funded by academic grants, we had assumed that our project and open sharing of our network would not be an issue. However, upon releasing a preliminary version of Hetionet (*Himmelstein and Jensen, 2015u*), a community reviewer informed us of legal barriers to integrating public data. In essence, both copyright (rights of exclusivity automatically granted to original works) and terms of use (rules that users must agree to in order to use a resource) place legally binding restrictions on data reuse. In short, public data is not by default open data.

Hetionet v1.0 integrates 29 resources (*Table 4*), but two resources were removed prior to the v1.0 release. Of the total 31 resources (*Himmelstein et al., 2015i*), 5 were United States government works not subject to copyright, and 12 had licenses that met the Open Definition of knowledge version 2.1. Four resources allowed only non-commercial reuse. Most problematic were the remaining nine resources that had no license — which equates to all rights reserved by default and forbids reuse (*Oxenham, 2016*) — and one resource that explicitly forbid redistribution.

Additional difficulty resulted from license incompatibles across resources, which was caused primarily by non-commercial and share-alike stipulations. Furthermore, it was often unclear who owned the data (*Elliott, 2005*). Therefore, we sought input from legal experts and chronicled our progress (*Himmelstein et al., 2015i*; *Himmelstein, 2015k*; *Himmelstein et al., 2016r*; *Himmelstein, 2015a*; *Himmelstein, 2015d*).

Ultimately, we did not find an ideal solution. We had to choose between absolute compliance and Hetionet: strictly adhering to copyright and licensing arrangements would have decimated the

**Table 4.** The 29 public data resources integrated to construct Hetionet v1.0.

Components notes which types of nodes and edges in Hetionet v1.0 derived from the resource (as per the abbreviations in **Table 1 and 2**). Cat. notes the general category of license (**Himmelstein et al., 2015i**). Category 1 refers to United States government works that we deemed were not subject to copyright. Category 2 refers to resources with licenses that allow use, redistribution, and modification (although some restrictions may still exist). The subset of category 2 licenses that we deemed to meet the the Open Definition are denoted with $^{OD}$. Category 4 refers to resources without a license, hence with all rights reserved. References provides Research Resource Identifiers as well as citations to resource publications and related Project Rephetio materials. For information on license provenance, institutional affiliations, and funding for each resource, see the online table.

| Resource | Components | License | Cat. | References |
|---|---|---|---|---|
| Entrez Gene | G | custom | 1 | RRID:SCR_002473 (*Maglott et al., 2011*; *Himmelstein et al., 2015h*; *Himmelstein, 2016l*) |
| LabeledIn | CtD, CpD | custom | 1 | RRID:SCR_015667 (*Khare et al., 2014*; *Khare et al., 2015*; *Himmelstein and Khare, 2015s*) |
| MEDLINE | DlA, DpS, DrD | custom | 1 | RRID:SCR_002185 (*Himmelstein and Pankov, 2015a*; *Himmelstein, 2016u*) |
| MeSH | S | custom | 1 | RRID:SCR_004750 (*Himmelstein and Pankov, 2015a*; *Himmelstein, 2016h*) |
| Pathway Interaction Database | PW, GpPW | | 1 | RRID:SCR_006866 (*Schaefer et al., 2009*; *Pico and Himmelstein, 2015*; *Himmelstein and Pico, 2016a*) |
| Disease Ontology | D | CC BY 3.0 | 2$^{OD}$ | RRID:SCR_000476 (*Schriml et al., 2012*; *Kibbe et al., 2015*; *Himmelstein and Li, 2015d*; *Himmelstein, 2016g*) |
| DISEASES | DaG | CC BY 4.0 | 2$^{OD}$ | RRID:SCR_015664 (*Himmelstein and Jensen, 2015l*; *Himmelstein and Jensen, 2016c*; *Pletscher-Frankild et al., 2015*) |
| DrugCentral | PC, CbG, PCiC | CC BY 4.0 | 2$^{OD}$ | RRID:SCR_015663 (*Ursu et al., 2017*; *Himmelstein et al., 2016d*) |
| Gene Ontology | BP, CC, MF, GpBP, GpCC, GpMF | CC BY 4.0 | 2$^{OD}$ | RRID:SCR_002811 (*Ashburner et al., 2000*; *Huntley et al., 2015*; *Himmelstein et al., 2015g*; *Himmelstein et al., 2015f*) |
| GWAS Catalog | DaG | custom | 2$^{OD}$ | RRID:SCR_012745 (*Himmelstein and Baranzini, 2016b*; *MacArthur et al., 2017*; *Himmelstein, 2015h*; *Himmelstein et al., 2015v*) |
| Reactome | PW, GpPW | custom | 2$^{OD}$ | RRID:SCR_003485 (*Fabregat et al., 2016*; *Cerami et al., 2011*; *Pico and Himmelstein, 2015*; *Himmelstein and Pico, 2016a*) |
| LINCS L1000 | CdG, CuG, Gr > G | custom | 2$^{OD}$ | (*Himmelstein and Chung, 2015q*; *Himmelstein et al., 2016k*; *Himmelstein, 2015k*) |
| TISSUES | AeG | CC BY 4.0 | 2$^{OD}$ | RRID:SCR_015665 (*Santos et al., 2015*; *Himmelstein and Jensen, 2015g*; *Himmelstein and Jensen, 2015h*) |
| Uberon | A | CC BY 3.0 | 2$^{OD}$ | RRID:SCR_010668 (*Mungall et al., 2012*; *Malladi et al., 2015*; *Himmelstein, 2016m*) |
| WikiPathways | PW, GpPW | CC BY 3.0/custom | 2$^{OD}$ | RRID:SCR_002134 (*Kutmon et al., 2016*; *Pico et al., 2008*; *Pico and Himmelstein, 2015*; *Himmelstein and Pico, 2016a*) |
| BindingDB | CbG | mixed CC BY 3.0 and CC BY-SA 3.0 | 2$^{OD}$ | RRID:SCR_000390 (*Chen et al., 2001*; *Gilson et al., 2016*; *Himmelstein and Gilson, 2015i*; *Himmelstein et al., 2015d*) |
| DisGeNET | DaG | ODbL | 2$^{OD}$ | RRID:SCR_006178 (*Himmelstein, 2015f*; *Himmelstein and Piñero, 2016d*; *Piñero et al., 2015*; *Piñero et al., 2017*) |
| DrugBank | C, CbG, CrC | custom | 2 | RRID:SCR_002700 (*Law et al., 2014*; *Himmelstein, 2015b*; *Himmelstein, 2016i*; *Himmelstein et al., 2016r*) |
| MEDI | CtD, CpD | CC BY-NC-SA 3.0 | 2 | RRID:SCR_015668 (*Himmelstein et al., 2015e*; *Wei et al., 2013*) |
| PREDICT | CtD, CpD | CC BY-NC-SA 3.0 | 2 | (*Gottlieb et al., 2011*; *Himmelstein et al., 2015e*) |
| SIDER | SE, CcSE | CC BY-NC-SA 4.0 | 2 | RRID:SCR_004321 (*Kuhn et al., 2016*; *Himmelstein, 2015c*; *Himmelstein, 2016j*) |
| Bgee | AeG, AdG, AuG | | 4 | RRID:SCR_002028 (*Himmelstein et al., 2016f*; *Himmelstein and Bastian, 2015e*; *Himmelstein and Bastian, 2015f*; *Bastian et al., 2008*) |
| DOAF | DaG | | 4 | RRID:SCR_015666 (*Himmelstein, 2015g*; *Himmelstein, 2016s*; *Xu et al., 2012*) |
| ehrlink | CtD, CpD | | 4 | (*McCoy et al., 2012*; *Himmelstein, 2015j*) |
| Evolutionary Rate Covariation | GcG | | 4 | RRID:SCR_015669 (*Priedigkeit et al., 2015*; *Himmelstein and Partha, 2015r*; *Himmelstein, 2016w*) |
| hetio-dag | GiG | | 4 | (*Himmelstein and Baranzini, 2015a*; *Himmelstein et al., 2015z*; *Himmelstein and Baranzini, 2016e*) |
| Incomplete Interactome | GiG | | 4 | (*Himmelstein et al., 2015z*; *Himmelstein and Baranzini, 2016e*; *Menche et al., 2015*; *Himmelstein, 2015a*) |

*Table 4 continued on next page*

*Table 4 continued*

| Resource | Components | License | Cat. | References |
|---|---|---|---|---|
| Human Interactome Database | GiG | | 4 | RRID:SCR_015670 (*Himmelstein et al., 2015z*; *Himmelstein and Baranzini, 2016e*; *Rual et al., 2005*; *Venkatesan et al., 2009*; *Yu et al., 2011*; *Rolland et al., 2014*) |
| STARGEO | DdG, DuG | | 4 | (*Himmelstein et al., 2015a*; *Himmelstein et al., 2016j*; *Hadley et al., 2017*) |

DOI: https://doi.org/10.7554/eLife.26726.011

network. On the other hand, in the United States, mere facts are not subject to copyright, and fair use doctrine helps protect reuse that is transformative and educational. Hence, we choose a path forward which balanced legal, normative, ethical, and scientific considerations.

If a resource was in the public domain, we licensed any derivatives as CC0 1.0. For resources licensed to allow reuse, redistribution, and modification, we transmitted their licenses as properties on the specific nodes and relationships in Hetionet v1.0. For all other resources — for example, resources without licenses or with licenses that forbid redistribution — we sent permission requests to their creators. The median time till first response to our permission requests was 16 days, with only two resources affirmatively granting us permission. We did not receive any responses asking us to remove a resource. However, we did voluntarily remove MSigDB (*Liberzon et al., 2011*), since its license was highly problematic (*Himmelstein, 2015d*). As a result of our experience, we recommend that publicly funded data should be explicitly dedicated to the public domain whenever possible.

## Permuted hetnets

From Hetionet, we derived five permuted hetnets (*Himmelstein, 2016b*). The permutations preserve node degree but eliminate edge specificity by employing an algorithm called XSwap to randomly swap edges (*Hanhijärvi et al., 2009*). To extend XSwap to hetnets (*Himmelstein and Baranzini, 2015a*), we permuted each metaedge separately, so that edges were only swapped with other edges of the same type. We adopted a Markov chain approach, whereby the first permuted hetnet was generated from Hetionet v1.0, the second permuted hetnet was generated from the first, and so on. For each metaedge, we assessed the percent of edges unchanged as the algorithm progressed to ensure that a sufficient number of swaps had been performed to randomize the network (*Himmelstein, 2016b*). Permuted hetnets are useful for computing the baseline performance of meaningless edges while preserving node degree (*Himmelstein, 2015l*). Since, our use of permutation focused on assessing Δ AUROC, a small number of permuted hetnets was sufficient, as the variability in a metapath's AUROC across the permuted hetnets was low.

## Graph databases and Neo4j

Traditional relational databases — such as SQLite, MySQL, and PostgreSQL — excel at storing highly structured data in tables. Connectivity between tables is accomplished using foreign-key references between columns. However, for many biomedical applications the connectivity between entities is of foremost importance. Furthermore, enforcing a rigid structure of what attributes an entity may possess is less important and often unnecessarily prohibitive. Graph databases focus instead on capturing connectivity (relationships) between entities (nodes). Accordingly, graph databases such as Neo4j offer greater ease when modeling biomedical relationships and superior performance when traversing many levels of connectivity (*Yoon et al., 2017*; *Jaiswal, 2013*). Until recently, graph database adoption in bioinformatics was limited (*Have and Jensen, 2013*). However lately, the demand to model and capture biological connectivity at scale has led to increasing adoption (*Lysenko et al., 2016*; *Balaur et al., 2016*; *Summer et al., 2016*; *Mungall et al., 2017*).

We used the Neo4j graph database for storing and operating on Hetionet and noticed major benefits from tapping into this large open source ecosystem (*Himmelstein, 2015m*). Persistent storage with immediate access and the Cypher query language — a sort of SQL for hetnets — were two of the biggest benefits. To facilitate our migration to Neo4j, we updated hetio — our existing Python package for hetnets (*Himmelstein, 2016g*) — to export networks into Neo4j and DWPC queries to Cypher. In addition, we created an interactive GraphGist for Project Rephetio, which introduces our approach and showcases its Cypher queries. Finally, we created a public Neo4j

instance (*Himmelstein, 2016i*), which leverages several modern technologies such Neo4j Browser guides, cloud hosting with HTTPS, and Docker deployment (*Belmann et al., 2015*; *Beaulieu-Jones and Greene, 2017*).

## Machine learning approach

Project Rephetio relied on the previously published DWPC metric to generate features for compound–disease pairs. The DWPC measures the prevalence of a given metapath between a given source and target node (*Himmelstein and Baranzini, 2015a*). It is calculated by first extracting all paths from the source to target node that follow the specified metapath. Next, each path is weighted by taking the product of the node degrees along the path raised to a negative exponent. This damping exponent — the sole parameter — thereby determines the extent that paths through high-degree nodes are downweighted: we chose $w = 0.4$ based on our past optimizations (*Himmelstein and Baranzini, 2015a*). The DWPC equals the sum of the path weights (referred to as path-degree products). Traversing the hetnet to extract all paths between a source and target node, which we performed in Neo4j, is the most computationally intensive step in computing DWPCs (*Himmelstein and Lizee, 2016t*). For future work, we are exploring matrix multiplication approaches, which could improve runtime several orders of magnitude.

Project Rephetio made several refinements to metapath-based hetnet edge prediction compared to previous studies (*Himmelstein and Baranzini, 2015a*; *Sun et al., 2011*). First, we transformed DWPCs by mean scaling and then taking the inverse hyperbolic sine (*Burbidge et al., 1988*) to make them more amenable to modeling (*Himmelstein et al., 2016s*). Second, we bifurcated the workflow into an all-features stage and an all-observations stage (*Himmelstein, 2016k*). The all-features stage assesses feature performance and does not require computing features for all negatives. Here, we selected a random subset of 3020 (4 × 755) negatives. Little error was introduced by this optimization, since the predominant limitation to performance assessment was the small number of positives (755) rather than negatives. Based on the all-features performance assessment (*Himmelstein, 2015n*), we selected 142 DWPCs to compute on all observations (all 209,168 compound–disease pairs). The feature selection was designed to remove uninformative features (according to permutation) and guard against edge-dropout contamination (*Himmelstein, 2016h*). Third, we included 14 degree features, which assess the degree of a specific metaedge for either the source compound or target disease.

## Network support of predictions

To improve the interpretability of the predictions, we developed a method for decomposing a prediction into its network support (*Himmelstein, 2016e*). This information is deployed to our Neo4j Browser guides, allowing users to assess the biomedical evidence contributing to a given prediction. First, we used logistic regression terms to quantify the contribution of metapaths that positively support a prediction. Second, we decomposed a metapath's contribution, according to its DWPC, into specific paths contributions. Finally, we aggregated paths based on their source (first) or target (last) edge to quantify the contribution of specific edges of the source compound or target disease (*Himmelstein, 2016f*).

Using the acamprosate–epilepsy prediction as an example, we first quantified metapath contributions: 40% of the prediction was supported by *CbGbCtD* paths, 36% by *CbGaD* paths, 11% by *CcSEcCtD* paths, 8% by *CbGpPWpGaD* paths, and 5% by *CbGeAlD* paths. Second, we calculated path contributions: *Acamprosate–binds–GRM5–associates–epilepsy syndrome* was the most supportive path, contributing 11% of the prediction. Finally, we aggregated path contributions to calculate that the source edge of *Acamprosate—binds—GRM5* contributed 23% of the prediction, while the target edge of *epilepsy syndrome–treats–Felbamate* contributed 12%.

## Prior probability of treatment

The 755 treatments in Hetionet v1.0 are not evenly distributed between all compounds and diseases. For example, methotrexate treats 19 diseases and hypertension is treated by 68 compounds. We estimated a prior probability of treatment — based only on the treatment degree of the source compound and target disease — on 744,975 permutations of the bipartite treatment network (*Lizee and*

*Himmelstein, 2016a*). Methotrexate received a 79.6% prior probability of treating hypertension, whereas a compound and disease that both had only one treatment received a prior of 0.12%.

Across the 209,168 compound–disease pairs, the prior predicted the known treatments with AUROC = 97.9%. The strength of this association threatened to dominate our predictions. However, not modeling the prior can lead to omitted-variable bias and confounded proxy variables. To address the issue, we included the logit-transformed prior, without any regularization, as a term in the model. This restricted model fitting to the 29,799 observations with a nonzero prior — corresponding to the 387 compounds and 77 diseases with at least one treatment. To enable predictions for all 209,168 observations, we set the prior for each compound–disease pair to the overall prevalence of positives (0.36%).

This method succeeded at accommodating the treatment degrees. The prior probabilities performed poorly on the validation sets with AUROC = 54.1% on DrugCentral indications and AUROC = 62.5% on clinical trials. This performance dropoff compared to training shows the danger of encoding treatment degree into predictions. The benefits of our solution are highlighted by the superior validation performance of our predictions compared to the prior (*Figure 3*).

## Indication sets

We evaluated our predictions on four sets of indications as shown in *Figure 3*.

- *Disease Modifying* — the 755 disease-modifying treatments in PharmacotherapyDB v1.0. These indications are included in the hetnet as *treats* edges and used to train the logistic regression model. Due to edge dropout contamination and self-testing (*Himmelstein, 2016h*; *Lizee and Himmelstein, 2016b*), overfitting could potentially inflate performance on this set. Therefore, for the three remaining indication sets, we removed any observations that were positives in this set.
- *DrugCentral* — We discovered the DrugCentral database after completing our physician curation for PharmacotherapyDB. This database contained 210 additional indications (*Himmelstein et al., 2016d*). While we didn't curate these indications, we observed a high proportion of disease-modifying therapy.
- *Clinical Trial* — We compiled indications that have been investigated by clinical trial from ClinicalTrials.gov (*Himmelstein, 2016d*). This set contains 5594 indications. Since these indications were not manually curated and clinical trials often show a lack of efficacy, we expected lower performance on this set.
- *Symptomatic* — 390 symptomatic indications from PharacotherapyDB. These edges are included in the hetnet as *palliates* edges.

Only the Clinical Trial and DrugCentral indication sets were used for external validation, since the Disease Modifying and Symptomatic indications were included in the hetnet. As an aside, several additional indication catalogs have recently been published, which future studies may want to also consider (*Himmelstein et al., 2015e*; *Brown and Patel, 2017*; *Shameer et al., 2017*; *Sharp, 2017*).

## Realtime open science and thinklab

We conducted our study using Thinklab — a platform for real-time open collaborative science — on which this study was the first project (*Himmelstein et al., 2015c*). We began the study by publicly proposing the idea and inviting discussion (*Himmelstein et al., 2015k*). We continued by chronicling our progress via discussions. We used Thinklab as the frontend to coordinate and report our analyses and GitHub as the backend to host our code, data, and notebooks. On top of our Thinklab team consisting of core contributors, we welcomed community contribution and review. In areas where our expertise was lacking or advice would be helpful, we sought input from domain experts and encouraged them to respond on Thinklab where their comments would be CC BY licensed and their contribution rated and rewarded.

In total, 40 non-team members commented across 86 discussions, which generated 622 comments and 191 notes (*Figure 6*). Thinklab content for this project totaled 145,771 words or 918,837 characters (*Himmelstein and Lizee, 2016v*). Using an estimated 7000 words per academic publication as a benchmark, Project Rephetio generated written content comparable in volume to 20.8 publications prior to its completion. We noticed several other benefits from using Thinklab including forging a community of contributors (*Patil and Siegel, 2009*); receiving feedback during the early stages when feedback was most actionable (*Mietchen et al., 2015*); disseminating our research

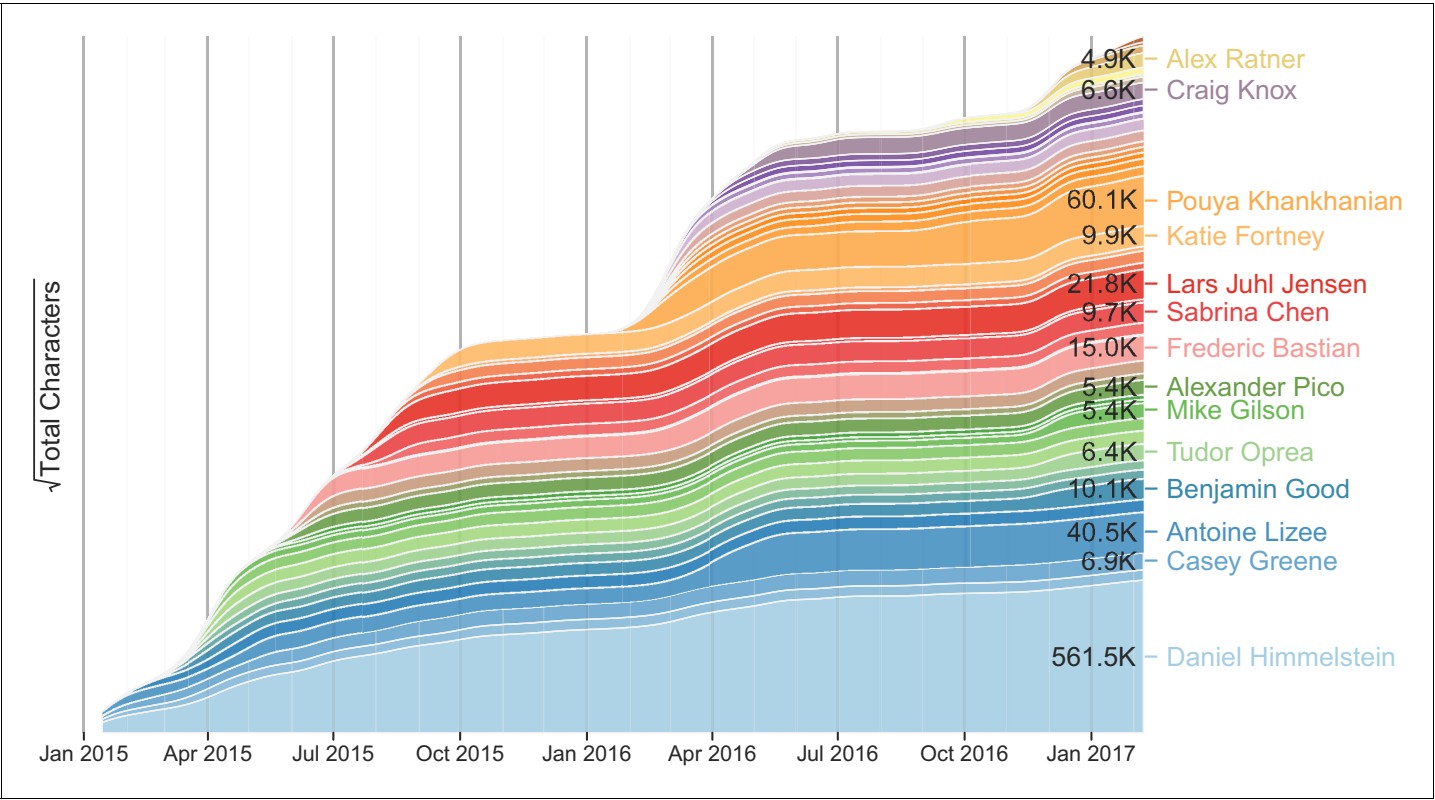

**Figure 6.** The growth the Project Rephetio corpus on Thinklab over time. This figure shows Project Rephetio contributions by user over time. Each band represented the cumulative contribution of a Thinklab user to discussions in Project Rephetio (***Himmelstein and Lizee, 2016v***). Users are ordered by date of first contribution. Users who contributed over 4500 characters are named. The square root transformation of characters written per user accentuates the activity of new contributors, thereby emphasizing collaboration and diverse input.
DOI: https://doi.org/10.7554/eLife.26726.012

without delay (***Powell, 2016***; ***Vale, 2015***); opening avenues for external input (***Allison et al., 2016***); facilitating problem-oriented teaching (***Himmelstein et al., 2016t***; ***Waldrop, 2015***); and improving our documentation by maintaining a publication-grade digital lab notebook (***Giles, 2012***).

Thinklab began winding down operations in July 2017 and has switched to a static state. While users will no longer be able to add comments, the corpus of content remains browsable at https://think-lab.github.io and available in machine-readable formats at dhimmel/thinklytics.

The preprint for this study is available at doi.org/bs4f (***Himmelstein et al., 2016u***). The manuscript was written in markdown, originally on Thinklab at doi.org/bszr (***Himmelstein et al., 2016v***). In August 2017, we switched to using the Manubot system to generate the manuscript. With Manubot, a GitHub repository (dhimmel/rephetio-manuscript) tracks the manuscript's source code, while continuous integration automatically rebuilds the manuscript upon changes. As a result, the latest version of the manuscript is always available at dhimmel.github.io/rephetio-manuscript. Additionally, readers can leave feedback or questions for the Project Rephetio team via GitHub Issues.

## Software and data availability

All software and datasets from Project Rephetio are publicly available on GitHub, Zenodo, or Figshare (***Himmelstein et al., 2017b***). Additional documentation for these materials is available in the corresponding Thinklab discussions. For reader convenience, software, datasets, and Thinklab discussions have been cited throughout the manuscript as relevant. Copies of the most relevant Github repositories are archived at: https://github.com/elifesciences-publications/hetionet; https://github.com/elifesciences-publications/integrate; https://github.com/elifesciences-publications/learn; https://github.com/elifesciences-publications/hetio and https://github.com/elifesciences-publications/rephetio-manuscript.

## Acknowledgements

We are immensely grateful to our Thinklab contributors who joined us in our experiment of radically open science. The following non-team members provided contributions that received five or more Thinklab points: Lars Juhl Jensen, Frederic Bastian, Alexander Pico, Casey Greene, Benjamin Good, Craig Knox, Mike Gilson, Chris Mungall, Katie Fortney, Venkat Malladi, Tudor Oprea, MacKenzie Smith, Caty Chung, Allison McCoy, Alexey Strokach, Ritu Khare, Greg Way, Marina Sirota, Ragha-vendran Partha, Oleg Ursu, Jesse Spaulding, Gaya Nadarajan, Alex Ratner, Scooter Morris, Alessan-dro Didonna, Alex Pankov, Tong Shu Li, and Janet Piñero. Additionally, the founder of Thinklab, Jesse Spaulding, supported community contributions and developed the platform with Project Rephetio's needs in mind. We also appreciate DigitalOcean's sponsorship the Hetionet Browser to cover its hosting costs. Finally, we would like to thank Neo Technology, whose staff provided excellent technical support. This material is based upon work supported by the National Science Foundation Graduate Research Fellowship under Grant Number 1144247 to DSH. SEB is supported by NINDS/NIH grant number 5R01NS088155 and the Heidrich Family and Friends Foundation. DH is supported by the the National Cancer Institute of the National Institutes of Health under Award Number UH2CA203792 and the National Library of Medicine under Award Number 1U01LM012675. The content is solely the responsibility of the authors and does not necessarily represent the official views of the NIH.

## Additional information

### Funding

| Funder | Grant reference number | Author |
| --- | --- | --- |
| National Science Foundation | 1144247 | Daniel Scott Himmelstein |
| Heidrich Family and Friends Foundation | | Sergio E Baranzini |
| National Institutes of Health | 5R01NS088155 | Sergio E Baranzini |
| National Cancer Institute | UH2CA203792 | Dexter Hadley |
| U.S. National Library of Medi-cine | 1U01LM012675 | Dexter Hadley |

The funders had no role in study design, data collection and interpretation, or the decision to submit the work for publication.

### Author contributions

Daniel Scott Himmelstein, Conceptualization, Data curation, Software, Formal analysis, Validation, Investigation, Visualization, Methodology, Writing—original draft, Project administration; Antoine Lizee, Software, Formal analysis, Visualization, Methodology; Christine Hessler, Supervision, Validation, Methodology; Leo Brueggeman, Resources, Data curation, Software, Formal analysis, Methodology; Sabrina L Chen, Data curation, Formal analysis, Validation; Dexter Hadley, Resources, Data curation, Software, Formal analysis; Ari Green, Conceptualization, Validation; Pouya Khankhanian, Conceptualization, Data curation, Software, Formal analysis; Sergio E Baranzini, Conceptualization, Resources, Supervision, Funding acquisition, Investigation, Visualization, Methodology, Writing—review and editing

### Author ORCIDs

Daniel Scott Himmelstein [iD] http://orcid.org/0000-0002-3012-7446
Sergio E Baranzini [iD] http://orcid.org/0000-0003-0067-194X

### Decision letter and Author response

Decision letter https://doi.org/10.7554/eLife.26726.016
Author response https://doi.org/10.7554/eLife.26726.017

# Additional files

## Supplementary files
• Transparent reporting form
DOI: https://doi.org/10.7554/eLife.26726.013

## Major datasets
The following previously published dataset was used:

| Author(s) | Year | Dataset title | Dataset URL | Database, license, and accessibility information |
|---|---|---|---|---|
| Himmelstein D, Brueggeman L, Baranzini S | 2017 | Figshare depositions from Project Rephetio | https://doi.org/10.6084/m9.figshare.c.2861359.v1 | Available at figshare under a CC0 Public Domain licence |

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
