## [Decision Letter]

Thank you for submitting your article "Systematic integration of biomedical knowledge prioritizes drugs for repurposing" for consideration by *eLife*. Your article has been reviewed by three peer reviewers, and the evaluation has been overseen by a Reviewing Editor and Aviv Regev as the Senior Editor. The following individuals involved in review of your submission have agreed to reveal their identity: Jason Moore (Reviewer #1); Amitabh Sharma (Reviewer #3).

The reviewers have discussed the reviews with one another and the Reviewing Editor has drafted this decision to help you prepare a revised submission.

This paper describes a very large resource of information on potential drug target interaction and its use for the prediction of drug repositioning opportunities The large network ensembles 47k nodes (11 kinds) and 2.25M edges (24 kinds) from 29 public databases. The network provides new relations that could lead to interesting discoveries. The algorithmic proposal is simple and reasonable, the testing makes sense and it is based on independent data sets. The examples provided are interesting as illustration of the best results, i.e., insights into epilepsy and nicotine dependence treatment.

There are a few points that require careful attention:

Justifications for the cutoffs used to define edges. Edges have a different level of reliability ranging very differently depending on the considered domain (ex. gene expression level). It is not clear how this has been taken into account/normalised. How edge reliability has been made comparable across resources and data types? Additionally, the authors have used different discretization threshold for different information domains. It is not clear how these thresholds have been determined and how/if they could potentially impact the predictive ability of the presented approach. This should be explained and each choice on this regard numerically justified.

The test sets used to validate the predictive ability of the Rephetio approach look strongly unbalanced toward negative cases (in 3 cases out of 4 the number of negatives is 3 orders of magnitude larger than the positives). It is necessary to assess the impact performance evaluation comparing with balanced sets of negatives and positives.

Other comments below are related with areas that require better explanations, clarification and editing:

- It was a bit confusing to map the nodes mentioned in the Introduction and Results to the Materials and methods. For example, anatomy is not mentioned in the nodes section of the Materials and methods. Making this more consistent might help the reader.

- It might be useful to expand some of your text on graph databases since they are very new and not many people know what they are or why they are useful.

- Edges of Hetionet are directed/undirected or a mixture of both. This inherently affects the definition of a path.

- The analysis of pathways of at most length 4 should be justified. What is the average length of a path between any node-pairs? And how loops are managed in this regard?

- The selection of the cases (e.g. methapaths Figure 3) should make clear how many good cases are there by for example providing a table with the best performing ones.

- The labels in Figure 2 are annoyingly all abbreviated. To improve readability I would suggest explicitly indicating the meta edge type.

- Even if referencing their relevant previous publication the "degree-weighted path count" should be briefly explained. Particularly, how this algorithm penalizes the paths involving high degree nodes?

- The multiple sclerosis example mentioned in the third paragraph of the subsection “Hetionet v1.0”: are the results shown anywhere? I believe the 4 nodes mentioned gene, disease, BP and anatomy? What exactly are the 5 types of interactions (guess: GpBP, DaG, DdG, DuG, AuG)? I understand that the authors have to contend with only a few examples to demonstrate the functionalities of their tools but still, it would help the reader to visualize this example if these details were presented.

- Related to the above: It would help the reader to show the four lines of Cypher code in the MS example to demonstrate the ease of use claimed by the authors. Or alternatively, to refer the reader to the part of the website where the syntax is introduced. Without this, the sentence "Furthermore, the portion of the query to identify paths meeting the above specification required only four lines of Cypher code" is left up in the air a little bit.

- In the second paragraph of the subsection “Systematic mechanisms of efficacy”, could the authors elaborate on the DWPC delta AUROC? The concept has been introduced in a previous publication, but it would be helpful to recap the definition so as to induce more biological insight in the reader.

- In Figure 2, it took me a while to figure out that the dot sizes are not continuous but instead there are only two dot sizes with small dots corresponding to non-significant (FDR>0.05) and larger dots corresponding to significant (FDR<0.05) metapaths. It would help to put a small legend clarifying this.

- The selected metapaths in Table 3. We see that good (all of the 11 positive covariates of the following section) as well as poor predictors are both shown. According to what criteria were they selected? Is it just a random selection intended to show the range of parameters and what they signify? If so, this should be noted.

- "29,044 non-treatments": If non-treatments are any disease-compound pairs that are not the 755 known treatments, then the number should be much higher. It would be helpful if the authors refer the reader to the Materials and methods section "Prior probability of treatment" here.

- The permutation test has to be more clearly explained and potentially combined or compared with some previously published approaches (BiRewire for example.

- The baseline performances of their network the authors assembled 5 randomized versions of it. This number seems to be very small and this section requires clarification.

- Did the Thinklab contributors – pre-reviewing the paper – consent to having their names appear in a publication?

- The analysis described in the first Results section (Systematic mechanisms of efficacy) leads to the authors' conclusion that the frequency of the information types in the metapaths for selected existing drug-disease pairs is higher for those traditionally considered by pharmacology and it is particularly low for metaedges involving gene expression and transcriptional data. In the eye of the authors this should tune down the recent excitement surrounding this type of data. This argument looks a bit circular. The authors have used a set of established drug-disease possibly involving many approved drugs. Drug discovery pipelines have been typically guided so far by knowledge of disease mechanisms, chemical structures of drug candidates and targets. Should not be obvious that for approved drugs what lead to their development is reflected by the enriched meta edges found by the authors?

- Additionally, most of the recent excitement around the use of transcriptional data for drug repurposing comes from the development of signature matching tools exploiting drug-drug similarities and drug-disease anti-similarity at the level of transcriptional signatures (respectively elicited by the drugs under consideration upon treatment of in vitro models, and from contrasting diseased vs. normal state). Would this be worthy the inclusion of other two type of edges (transcriptional signature similarity/anti-similarity at the whole signature level)? This possibility should be at least discussed.

Finally, Heterogeneous network presented in this study unambiguously refers to "aggregated" networks consisting of the sum of multiple types of nodes and edges. While heterogeneous networks present perhaps the most viable way of integrating multiple and diverse types of biomedical data and are therefore very valuable to gaining integrated biological insights through machine learning approaches, it is important to distinguish them from multilayer networks, whose theory and mathematical framework has been established in the reviews mentioned. The current diversity of nomenclature in the filed stems from the fact that, at least for now, these slightly different types of multilayer networks have to distinguished for their mathematical treatment. On the other hand, no such mathematical framework exists for generic heterogeneous networks such as HetNet, but rather, an exhaustive survey of node/edge combinations (metapaths) is needed, such as the one presented in this paper. The authors should note the fundamental difference between multilayer and aggregated network approaches when making this comparison.

---

## [Author Response]

There are a few points that require careful attention:Justifications for the cutoffs used to define edges. Edges have a different level of reliability ranging very differently depending on the considered domain (ex. gene expression level). It is not clear how this has been taken into account/normalised. How edge reliability has been made comparable across resources and data types? Additionally, the authors have used different discretization threshold for different information domains. It is not clear how these thresholds have been determined and how/if they could potentially impact the predictive ability of the presented approach. This should be explained and each choice on this regard numerically justified.

We appreciate this reviewer’s viewpoint on the potentially different thresholds for each edge type and have made our best efforts to choose them rigorously. However, our approach does not require that each type of edge is equally reliable. For example, let’s assume 90% of *Gene-associates-Disease* edges are true whereas only 70% of *Compound-upregulates-Gene* edges are true. Such a disparity would potentially affect the performance of a specific metaedge, but since we use a supervised machine learning approach, the impact on predictions of a given edge type is dependent on its reliability.

Whenever selecting a threshold for a given edge type, our goal was to balance precision (percent of edges that are true) and recall (percent of real relationships that are included as edges). We attempted to pick thresholds so that we were reasonably confident in any given edge being true (high precision). In order to achieve moderate levels of recall, we generally selected more permissive thresholds for high-throughput data sources.

While constructing gold standards for each metaedge would be an interesting direction, it was outside of the scope of this study. Therefore, for most metaedges, we did not have objective measures of precision and recall rates at various inclusion thresholds. However, we did attempt to balance precision and recall when selecting inclusion thresholds, although this process was often qualitative rather than quantitative. In many cases, the final decision was reached after considerable discussion on Thinklab and exploratory data analyses. In addition, we often asked the authors of a particular resource to comment on an appropriate threshold.

Of course, this is not an entirely satisfactory answer. One intriguing (although application specific) solution would be a performance-driven approach. For example, optimizing an edge inclusion threshold to maximize performance. Since Hetionet v1.0 has many edge inclusion thresholds, difficult decisions would need to be made on whether to optimize the thresholds independently or jointly. The primary barrier we faced, and why we didn’t perform these analyses, was computational. Such an approach would add considerable code complexity by requiring a unified pipeline that interacts with many discrete repositories. In addition, we were already running up against compute time barriers, without performance-driven threshold optimization. We’re currently pursuing matrix multiplication approaches to compute DWPCs, which would render performance-driven edge threshold optimization computationally feasible. Therefore, we agree with the reviewers that this is an important future direction.

We recommend readers that are interested in our edge threshold determinations to refer to the cited Thinklab discussions and source code on GitHub. As a quick note for technical readers, the exact provenance for each relationship in Hetionet v1.0 can be deduced from working backwards from the integrate.ipynb notebook source code. Below we elaborate on several examples regarding edge threshold determination:

*Compound*–*binds*–*Gene* relationships from BindingDB. We discussed various techniques for consolidating multiple binding affinity experiments and selecting an edge inclusion threshold. Mike Gilson, the principal investigator behind BindingDB, commented: “If a given compound and protein target have multiple measurements of different types, I’d probably use them in the following order of preference: Kd over Ki over IC50.” In a subsequent comment, we describe the method we chose: “multiple affinities for the same bindingdb–uniprot pairs were resolved by preferentially selecting Kd over Ki over IC50 and taking a geometric mean when there were multiple measurements of the same measure.” We further explain our choice of binding affinity threshold, writing: “Setting an affinity threshold at 1 μM (1000 nanomolar) – suggested by both [Mike Gilson] and [Alessandro Didonna] – retained ~20% of interactions.”

Compound–resembles–Compound relationships. We generated chemical similarity relationships based on the Dice coefficient between extended connectivity fingerprints (ECFPs). Mike Keiser commented in reference to his past research:

“Using ECFP4 (i.e., Morgan with radius 2 in RDKit) and a Tanimoto coefficient, we found cutoffs more around 0.28 (the range can vary pretty substantially depending on fingerprint type used). In general, 0.5 is considered pretty high similarity for ECFP/Morgan fingerprints at least with Tanimoto coefficients (I’m less sure of the Dice coefficient equivalents, off-hand).”

After evaluating the distribution of pairwise similarity scores as well as examining how chemical similarity associated with transcriptional similarity and the prevalence of shared targets between compound pairs, we decided on a threshold of 0.5.

*Anatomy*–*expresses*–*Gene* relationships from TISSUES. We discussed how to consolidate evidence from multiple channels of evidence with TISSUES’ principal investigator Lars Juhl Jensen. Dr. Jensen wrote: “If you want to enforce a hard cutoff to make things binary, I would urge you to at least take the integrated scores that takes everything into account and apply the cutoff to that. In this case a score of 3 might be appropriate.” We ended up selecting a more stringent threshold of 2 based on the observed distribution of integrated scores (notebook Cell 3), which indicated that increasing the threshold from 2 to 3 would benefit recall only minimally.

*Anatomy*–*up/downregulates*–*Gene* relationships from Bgee. For a given gene–anatomy pair, Bgee provides several reliability measures for whether the gene is underexpressed or overexpressed. We included unambiguously “low quality” or “high quality” relationships, based on the advice of Frederic Bastian, a lead architect of Bgee, who wrote:

“I would definitely use the “low quality” as well here. Because the overall call generated is based on a voting system weighted by p-values, so even if it is “low quality” because of conflicting analyses, the best p-value has won anyway.”

We investigated the frequency of different call qualities (notebook Cell 14) and discussed the inclusion threshold options extensively.

*Gene–covaries–Gene* relationships from Evolutionary Rate Covariation. We selected a covariation threshold of 0.75. This determination was based on the relative frequency of positive to negative covariation coefficients. From this analysis, we reasoned that “selecting a threshold of ERC > 0.75 would lead to a false discovery rate of approximately 10%."

*Gene*–*associates*–*Disease* relationships from DisGeNET. We discussed the threshold with the resource’s creator Janet Piñero, positing that “we would like a permissive threshold, allowing up to a ~30% false discovery rate.” Dr. Piñero responded: “If you choose score ≥ 0.06, then you will be including associations reporting by curated sources, or having animal models supporting them, or being reported by several papers (20–200). It will not be permissive, though (less than 10% of GDAs satisfies this criteria).” Despite not being the most permissive, we went with the score ≥ 0.06 threshold, since we compiled gene–disease associations from three other resources, and therefore could afford to be more stringent for each given resource.

MEDLINE co-occurrence relationships (*Disease–localizes–Anatomy, Disease–presents–Symptom*, and *Disease–resembles–Disease*). We included co-occurring terms with *p* < 0.005 from a Fisher’s exact test assessing whether terms (MEDLINE topics) occurred more frequently together than expected by chance. We selected this threshold after examining the resulting datasets for each relationship type (DlA, DpS, DrD) and observing that this threshold appeared to give high precision with sufficient recall.

As the above examples illustrate, edge inclusion thresholds were chosen based on extensive empirical assessment, data exploration, and advice from resource creators and other experts. Finally, for many relationships, we include the confidence scores as an edge property, so Hetionet users can perform additional filtering according to their needs.

The test sets used to validate the predictive ability of the Rephetio approach look strongly unbalanced toward negative cases (in 3 cases out of 4 the number of negatives is 3 orders of magnitude larger than the positives). It is necessary to assess the impact performance evaluation comparing with balanced sets of negatives and positives.

We rely on AUROC as our primary measure of classifier performance since it is prevalence agnostic. Hence, Figure 3 are unaffected by the imbalance of positives and negatives. We acknowledge that both precision and recall are affected by the imbalance (Figure 3), however this is an important aspect of their utility. In the context of genomic predictions, Myers et al. 2006 go into more detail on this point, stating:

“To avoid such misleading evaluations, the balance of positives and negatives in the gold standard should match that of the application domain as closely as possible. Precision, or PPV, then becomes a direct, representative measure of how well one could expect a dataset or method to perform on whole-genome tasks.”

In our case, we included the full set of negatives in our various gold standards (validation sets), since we aimed to capture the unbalanced nature of drug efficacy. The point we are trying to highlight is that a random compound-disease pair has a low probability of being a treatment, and that’s an important aspect of the problem domain that should be reflected in our evaluation.

Other comments below are related with areas that require better explanations, clarification and editing:- It was a bit confusing to map the nodes mentioned in the Introduction and Results to the Materials and methods. For example, anatomy is not mentioned in the nodes section of the Materials and methods. Making this more consistent might help the reader.

The Nodes section of Materials and methods does document each node type. To help readers locate node descriptions, we have now bolded the first mention for each metanode in this section.

We did notice the Edges section of Materials and methods did not explicitly mention *Gene→regulates→Gene*, *Gene–participates–Pathway*, and *Pharmacologic Class–includes–Compound* edges, and has now been corrected. We thank the reviewer for catching this oversight.

- It might be useful to expand some of your text on graph databases since they are very new and not many people know what they are or why they are useful.

We have now updated the Neo4j section of the Materials and methods to “Graph databases and Neo4j”. This revised section contains a paragraph describing the utility of graph databases, especially in the historical context of relational databases. This section also provides a location for us to reference many of the exciting biomedical applications of Neo4j that have occurred concurrently with Project Rephetio.

- Edges of Hetionet are directed/undirected or a mixture of both. This inherently affects the definition of a path.

We added a Materials and methods section entitled “Directionality”, which aims to demonstrate how this is handled by Project Rephetio. This section distinguishes directionality from orientation, by exhibiting how undirected metaedges/edges can be traversed in an oriented manner (e.g. that *HADH–associates–obesity* and *obesity–associates–HADH* are the same undirected edge but in inverse orientations). In addition, we provide additional notes below.

Directionality is defined at the metaedge level. With the exception of *Gene→regulates→Gene*, all of Hetionet v1.0’s metaedges are undirected. In short, directionality is only ever necessary for metaedges that connect a metanode to itself. Note however that not all Gene–Gene metaedges are directed. For example, *Gene–interacts–Gene* and *Gene–covaries–Gene* are undirected, since there is no inherent directionality in the relationships they encode. Since the *Gene→regulates→Gene* metaedge is directed, a direction must be specified whenever an edge of that type is created in the hetio python package.

Hetio accounts for directionality, as needed, when traversing metapaths or paths. For example, the metapaths *CbGr>GaD* and *CbG<rGaD* are distinct. In the former, the first gene regulates the second gene. In the latter, the second gene regulates the first gene. However, since *Gene–interacts–Gene* edges are undirected, there is only one *CbGiGaD* metapath. When our network traversal infrastructure encounters a directed metaedge, only edges conforming to the specified direction are traversed. In the case of undirected metaedges, all of the corresponding edges are traversed, since there is no concept of directionality.

- The analysis of pathways of at most length 4 should be justified. What is the average length of a path between any node-pairs? And how loops are managed in this regard?

The limitation of metapaths at length 4 was a primarily computational decision. As seen in Figure 1, the number of possible metapaths grows combinatorially with increasing length. For Compound–Disease metapaths on Hetionet v1.0, there are 13 metapaths of length 2, 121 metapaths of length 3, and 1072 metapaths of length 4. Not only are longer metapaths more abundant, but computing each DWPC value for a longer metapath is more runtime intensive when using network traversal algorithms. The matrix multiplication methods currently under development will allow us to efficiently compute path counts and DWPCs for longer metapaths. However, we anticipate that the informativeness of DWPCs will diminish with excessively long metapaths.

Duplicate metanodes in a metapath are permitted. In other words, the metapaths we analyzed correspond to all *walks* on the metagraph from Compound to Disease with length 2–4. When calculating path counts and degree-weighted path counts, duplicate nodes are excluded.

- The selection of the cases (e.g. methapaths Figure 3) should make clear how many good cases are there by for example providing a table with the best performing ones.

Unfortunately, we are not sure what the reviewer means by “cases” in his/her comment. We thus interpreted this comment to mean either:

1) How many of the 209,168 compound–disease pairs received noteworthy predicted probabilities?

2) How many of 1,206 metapaths assessed were informative?

Regarding the first interpretation, the manuscript states: “Of the 3,980 predictions with a probability exceeding 1%, 586 corresponded to known disease-modifying indications, leaving 3,394 repurposing candidates.” Regarding the second interpretation, the manuscript states: “Overall, 709 of the 1,206 metapaths exhibited a statistically significant Δ AUROC at a false discovery rate cutoff of 5%.” For more information, we also refer readers to the interactive tables for predictions and metapaths online.

- The labels in Figure 2 are annoyingly all abbreviated. To improve readability I would suggest explicitly indicating the meta edge type.

We updated Figure 2 to use full metaedge names (commit). Regrettably, we still use abbreviations for metapaths in Figure 2 as their full names were too long.

- Even if referencing their relevant previous publication the "degree-weighted path count" should be briefly explained. Particularly, how this algorithm penalizes the paths involving high degree nodes?

We have now added a paragraph in Materials and methods, under “Machine learning approach”, that describes the DWPC metric in detail.

- The multiple sclerosis example mentioned in the third paragraph of the subsection “Hetionet v1.0”: are the results shown anywhere? I believe the 4 nodes mentioned gene, disease, BP and anatomy? What exactly are the 5 types of interactions (guess: GpBP, DaG, DdG, DuG, AuG)? I understand that the authors have to contend with only a few examples to demonstrate the functionalities of their tools but still, it would help the reader to visualize this example if these details were presented.

In response to this suggestion, we now include the Cypher code for the enhanced multiple sclerosis query within the manuscript. From the query, one can observe that the 4 node types are Disease, Gene, Biological Process, and Anatomy. Similarly the 5 relationship types are ASSOCIATES_DaG, INTERACTS_GiG, PARTICIPATES_GpBP, LOCALIZES_DlA, and UPREGULATES_AuG. Note that we adopted a slightly different nomenclature for Neo4j node labels and relationships types, as per Cypher style conventions and to help disambiguate metaedges.

- Related to the above: It would help the reader to show the four lines of Cypher code in the MS example to demonstrate the ease of use claimed by the authors. Or alternatively, to refer the reader to the part of the website where the syntax is introduced. Without this, the sentence "Furthermore, the portion of the query to identify paths meeting the above specification required only four lines of Cypher code" is left up in the air a little bit.

Thanks for this suggestion, which we have now incorporated this example into the manuscript.

- In the second paragraph of the subsection “Systematic mechanisms of efficacy”, could the authors elaborate on the DWPC delta AUROC? The concept has been introduced in a previous publication, but it would be helpful to recap the definition so as to induce more biological insight in the reader.

We have now added the following description: “A positive Δ AUROC indicates that paths of the given type tended to occur more frequently between treatments than non-treatments, after accounting for different levels of connectivity (node degrees) in the hetnet. In general terms, Δ AUROC assesses whether paths of a given type were informative of drug efficacy."

- In Figure 2, it took me a while to figure out that the dot sizes are not continuous but instead there are only two dot sizes with small dots corresponding to non-significant (FDR>0.05) and larger dots corresponding to significant (FDR<0.05) metapaths. It would help to put a small legend clarifying this.

We added a legend for dot size (commit), in addition to the sentence in the figure caption.

- The selected metapaths in Table 3. We see that good (all of the 11 positive covariates of the following section) as well as poor predictors are both shown. According to what criteria were they selected? Is it just a random selection intended to show the range of parameters and what they signify? If so, this should be noted.

Table 3 shows the metapaths that received a non-negligible positive coefficient in the logistic regression model as well as metapaths that are discussed as interesting findings. The full metapath table is available online. We have now clarified the caption.

- "29,044 non-treatments": If non-treatments are any disease-compound pairs that are not the 755 known treatments, then the number should be much higher. It would be helpful if the authors refer the reader to the Materials and methods section "Prior probability of treatment" here.

The 29,044 negatives used for training were all non-treatments between compounds and diseases that had at least one treatment. In other words, 29,044 negatives = 387 compounds × 77 diseases − 755 treatments. We now note in the Results section that “179,369 non-treatments were omitted as negative training observations because they had a prior probability of treatment equal to zero."

- The permutation test has to be more clearly explained and potentially combined or compared with some previously published approaches (BiRewire for example.

We appreciate the reviewers’ recommendation to look into BiRewire, which we were previously unaware of. Both BiRewire and our implementation apply the same underlying algorithm, which we refer to as XSwap as per Hanhijärvi et al. 2009. In both approaches, edge swaps (or “switching-steps") are successively applied to randomize edges while preserving node degree. A Markov chain approach begins each round of permutation from the last. For more information, see our detailed comparison online. In addition, the manuscript now describes the permutation methodology in greater detail.

- The baseline performances of their network the authors assembled 5 randomized versions of it. This number seems to be very small and this section requires clarification.

The primary purpose of the permuted hetnets was to compute the Δ AUROC of metapaths. This calculation takes the difference between the AUROC of a metapath and the average AUROC of the metapath on the 5 permuted hetnets. Given the large number of positives and negatives evaluated by the ROC curve, the AUROC values are highly stable. Therefore, 5 permutations was sufficient to calculate Δ AUROCs. In addition, we argue that the benefit to a much larger number of permutations would be small, since the noise inherent to the sole AUROC from the single unpermuted hetnet remains constant. We’ve updated the Permuted Hetnets section of the Methods to justify our choice.

- Did the Thinklab contributors – pre-reviewing the paper – consent to having their names appear in a publication?

Consent was not sought since *Thinklab* contributions were made publicly. Thinklab contributor names are listed in the acknowledgements and displayed in Figure 6. This is not considered as an endorsement of the study by the named individuals, but rather as a reflection of the publicly-available scientific record behind the study.

- The analysis described in the first Results section (Systematic mechanisms of efficacy) leads to the authors' conclusion that the frequency of the information types in the metapaths for selected existing drug-disease pairs is higher for those traditionally considered by pharmacology and it is particularly low for metaedges involving gene expression and transcriptional data. In the eye of the authors this should tune down the recent excitement surrounding this type of data. This argument looks a bit circular. The authors have used a set of established drug-disease possibly involving many approved drugs. Drug discovery pipelines have been typically guided so far by knowledge of disease mechanisms, chemical structures of drug candidates and targets. Should not be obvious that for approved drugs what lead to their development is reflected by the enriched meta edges found by the authors?

We acknowledge that our gold standard of treatments is biased towards traditional pharmacology and existing knowledge. However, we assume that known treatments are enriched for drug efficacy compared to random non-treatments (compound–disease pairs without known efficacy). Further, we hypothesize that high-throughput information types, if predictive of drug efficacy, would apply equally well to known and unknown treatments. Therefore, we would expect that metapaths based on high-throughput data sources (e.g. transcriptional edges) still prioritize known treatments if truly informative. Nonetheless, we agree that knowledge biases could favor the performance of metapaths based on traditional pharmacology. Accordingly we added the following sentences to the Discussion:

“One caveat is that omics-scale experimental data will likely play a larger role in developing the next generation of pharmacotherapies. Hence, were performance reevaluated on treatments discovered in the forthcoming decades, the predictive ability of these data types may rise.”

- Additionally, most of the recent excitement around the use of transcriptional data for drug repurposing comes from the development of signature matching tools exploiting drug-drug similarities and drug-disease anti-similarity at the level of transcriptional signatures (respectively elicited by the drugs under consideration upon treatment of in vitro models, and from contrasting diseased vs. normal state). Would this be worthy the inclusion of other two type of edges (transcriptional signature similarity/anti-similarity at the whole signature level)? This possibility should be at least discussed.

Regarding transcription-based methods for repurposing, our findings provide little support for “drug-disease anti-similarity” approaches, but do provide stronger support “drug-drug similarity” approaches. The “drug-disease anti-similarity” approach is evaluated by the *CuGdD* and *CdGuD* metapaths, whose respective Δ AUROCs were 1.1% and 1.7% (Table 3). The “drug-drug similarity” approach is evaluated by the *CuGuCtD* and *CdGdCtD* metapaths, whose respective Δ AUROCs were 4.4% and 3.8%. We do think that our current Discussion section accurately portrays these findings.

Not including metaedges for transcriptional (anti-)similarity at the whole signature level was a design decision motivated by the desire to avoid encoding duplicate information into multiple metaedges. For example, the hypothetical metaedge of *Compound–transcriptionally-resembles–Compound (CtrC*) would replicate information already provided by *Compound–upregulates–Gene* and *Compound–downregulates–Gene* edges. However, if we only included *CtrC* edges and not *CuG/CdG* edges, we could not assess metapaths such as CuGdD. But including *CuG/CdG* edges did allow us to assess *CtrC* relationships, just through longer metapaths that substitute *CuGuC* or *CdGdC* for *CtrC*.

Whether to collapse transcriptional signatures into similarity relationships likely depends on the application. The benefits of collapsing are lower dimensionality, reduced computational complexity, and fine-tuning the similarity metric. In certain cases, the decision was obvious in favor of collapsing. For example, the *Compound–resembles–Gene* relationships were calculated from chemical fingerprints that are composed of many bits representing molecular features. Here, we took a Dice coefficient between two compounds to create a similarity score, rather than having “molecular feature” nodes in the hetnet. This made sense since the only purpose of the molecular features was to assess compound resemblance. In addition, the chemical substructures encoded by the molecular features are computational abstractions rather than true biochemical entities.

Going forward, similar design decisions will be relevant. For example, morphological profiling can assess how a compound affects cell morphology (e.g. “size, shape, texture, intensity"). Since the morphological features are machine-generated and rather obscure in meaning, it likely makes the most sense to create *Compound-morphologically-resembles-Compound* relationships type. Of course, as more research becomes available on the utility of morphological features, the decision on how to encode these features in a hetnet can rely more heavily on the findings of prior work.

Finally, Heterogeneous network presented in this study unambiguously refers to "aggregated" networks consisting of the sum of multiple types of nodes and edges. While heterogeneous networks present perhaps the most viable way of integrating multiple and diverse types of biomedical data and are therefore very valuable to gaining integrated biological insights through machine learning approaches, it is important to distinguish them from multilayer networks, whose theory and mathematical framework has been established in the reviews mentioned. The current diversity of nomenclature in the filed stems from the fact that, at least for now, these slightly different types of multilayer networks have to distinguished for their mathematical treatment. On the other hand, no such mathematical framework exists for generic heterogeneous networks such as HetNet, but rather, an exhaustive survey of node/edge combinations (metapaths) is needed, such as the one presented in this paper. The authors should note the fundamental difference between multilayer and aggregated network approaches when making this comparison.

We appreciate the reviewer’s insights on the nuances of different graph conceptualizations. We agree that the term “aggregated” is helpful for explaining hetnets, and updated an Introduction sentence:

“Hetnets (short for heterogeneous networks) are networks with multiple types of nodes and relationships. They offer an intuitive, versatile, and powerful structure for data integration by aggregating graphs for each relationship type onto common nodes.”

Furthermore, we updated the Discussion to better reflect the “fundamental difference between multilayer and aggregated network approaches” (citations removed below):

“A 2014 analysis identified 78 studies using multilayer networks — a superset of hetnets (heterogeneous information networks) with the potential for additional dimensions, such as time. […] Compared to the existing mathematical frameworks for multilayer networks that must deal with layers other than type (such as the aspect of time), the primary obligation of hetnet algorithms is to be type aware.”